# Memory Augmented Optimizers for Deep Learning

Paul-Aymeric McRae [†1], Prasanna Parthasarathi [†1,2], Mahmoud Assran[1,2], and Sarath Chandar[1,3,4]

[1]Mila - Quebec AI Institute, Canada
[2]McGill University, Canada
[3]École Polytechnique de Montréal, Canada
[4]Canada CIFAR AI Chair

## Abstract

Popular approaches for minimizing loss in data-driven learning often involve an abstraction or an explicit retention of the history of gradients for efficient parameter updates. The aggregated history of gradients nudges the parameter updates in the right direction even when the gradients at any given step are not informative. Although the history of gradients summarized in meta-parameters or explicitly stored in memory has been shown effective in theory and practice, the question of whether *all* or only a subset of the gradients in the history are sufficient in deciding the parameter updates remains unanswered. In this paper, we propose a framework of memory-augmented gradient descent optimizers that retain a limited view of their gradient history in their internal memory. Such optimizers scale well to large real-life datasets, and our experiments show that the memory augmented extensions of standard optimizers enjoy accelerated convergence and improved performance on a majority of computer vision and language tasks that we considered. Additionally, we prove that the proposed class of optimizers with fixed-size memory converge under assumptions of strong convexity, regardless of which gradients are selected or how they are linearly combined to form the update step.

## 1 Introduction

Gradient-based learning involves minimizing a scalar-valued function $F(\theta)$ with respect to a parameter vector $\theta \in \mathbb{R}^d$ using an iterative procedure. When training over $n$ examples of input-output pairs, $(x_i, y_i)$, the optimization problem boils down to solving $\min_\theta F(\theta)$, where

$$F(\theta) = \frac{1}{n} \sum_{i=1}^{n} L_i(\theta), \quad \text{with} \quad L_i(\boldsymbol{\theta}) = \mathcal{L}\left(M_\theta(\boldsymbol{x}_i), y_i\right),$$

for some problem-dependent loss function $\mathcal{L}$ and a predictive model $M$ parameterized by $\theta$.

Stochastic Gradient Descent (SGD) (Robbins and Monro, 1951) is one common method used to tackle this problem, and is often preferred to full-batch Gradient Descent when the quantity of data required to train $\theta$ is large, since it can be more efficient to measure a single component of the gradient (or a mini-batch of gradients (Bottou, 1999)), and move in a noisy direction, than to compute a full gradient at each time step. Several techniques have been proposed to further accelerate the convergence of SGD (Ghadimi and Lan, 2016; Allen-Zhu and Yuan, 2016; Zhou et al., 2018). These include approaches that maintain a knowledge of previous gradients implicitly by summarizing them in a momentum buffer (Polyak, 1964b), and potentially adapting the learning rate based on the gradient statistics (Duchi et al., 2011; Hinton et al., 2012; Zeiler, 2012; Kingma and Ba, 2015).

Optimization techniques such as SGD with Momentum (Polyak, 1964a), AdaGrad (Duchi et al., 2011), RMSprop (Hinton et al., 2012), AdaDelta (Zeiler, 2012), and Adam (Kingma and Ba, 2015) maintain a set of buffers that track running moments of the gradients. Such light-weight techniques

---

†The two authors contributed equally to this paper.

have shown significant application advantage in practice, whereas in theory, algorithms that store all of the gradients, like SAG (Roux et al., 2012) and SAGA (Defazio et al., 2014), can achieve better convergence. A drawback of full-history methods is that the memory requirement linearly increases with the size of the data.

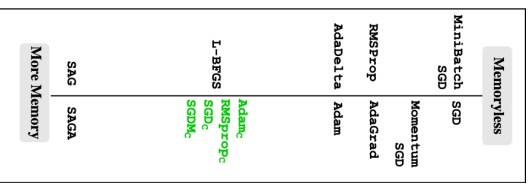

Figure 1: Common optimization algorithms placed over the spectrum of memory requirements. The proposed algorithms the $C$ variants have a fixed sized memory to store gradients during training.

The spectrum of gradient-based approaches (Figure 1) that use a knowledge of the past gradients has Adam, RMSprop and SGD with Momentum on the one end and approaches like SAG and SAGA on the other. An optimization algorithm in the middle of this spectrum can use less memory than full-history methods while providing richer updates than adaptive-gradient methods.

Limited-Memory BFGS (Nocedal, 1980) and its online, stochastic form, oLBFGS (Schraudolph et al., 2007) aim to exploit this trade-off, but may not converge faster than SGD (Mokhtari et al., 2015), and are thus disadvantaged compared to accelerated SGD variants. As a stepping stone towards reaping the advantages of both ends of the spectrum, we propose using memory to augment conventional optimizers. This memory is used to store a small set of *critical gradients* (e.g., gradients with a larger $\ell_2$-norm) that are occasionally updated with newer gradients. Specifically, rather than storing gradients for all the examples in the dataset, our proposed approach aims to find a smaller set of gradients to store that will aid in the optimization process. Through extensive experiments we show that this marginal increase in the memory requirement (Figure 1) provides accelerated convergence and even improved test performances in majority of the experiments.

In this work, we:

- Present a framework of memory-augmented optimizers compatible with popular gradient-based algorithms.
- Prove theoretically that such algorithms converge for smooth strongly convex objectives.
- Show that the proposed memory augmented optimizers can lead to faster convergence and better final performance through exhaustive empirical study on eight different architectures among four different tasks (classification, language modeling, natural language inference, and dialogue) and six different datasets.
- Demonstrate the memory augmented optimizers' robustness to gradient selection, replacement , or the aggregation techniques to summarize information in the memory.

## 2 MEMORY-AUGMENTED OPTIMIZERS

In this work, we propose augmenting standard optimizers with a fixed size memory of past gradients. The gradients are stored in the limited memory only when they are deemed *critical* as defined by their $\ell_2$-norm.

**Gradient Selection to the Memory** Letting $g_t := \nabla_\theta L_i(\theta_t)$ denote a component (or mini-batch) gradient at time-step $t$, we use $\|g_t\|_2$ as a scalar indicator of the importance of the gradient, which serves as a proxy for the priority of a gradient to remain in the memory buffer $g_c$. In order to ensure that the buffer eventually flushes out stale gradients, the proxy norms for gradients in the buffer are scaled down by a hyperparameter decay factor, denoted decay $\in [0, 1)$.

The proposed approach maintains a gradient buffer of fixed capacity $C$ and stores the gradients selected by a chosen heuristic. We refer our heuristic as the *critical gradients*, which stores the *top C* gradients by using the $\ell_2$-norm of $g_t$ to determine priority. Gradients in the buffer are stored as tuples $(\|g_t\|_2, g_t)$, akin to a key-value pair in a dictionary structure, where the key $\|g_t\|_2$ is referred to as the proxy norm, and decayed at each time step $t$ by a decay factor decay $\in [0, 1)$. At any iteration $t$ during training, the gradient in a full capacity priority buffer with the smallest $\ell_2$ proxy-norm is

replaced by the current mini-batch gradient $g_t$ if $\|g_t\|_2$ is greater than the smallest $\ell_2$ proxy-norm in the buffer. The multiplicative decay factor ensures that the buffer is frequently refreshed with more recent gradients. Note that only the proxy-norm used for the heuristic rule is decayed by the decay factor decay $\in [0, 1)$, while the gradient itself is not affected. This technique of storing the critical gradients is general enough to be employed in any deep learning model, and can be easily be combined with many existing adaptive gradient optimization algorithms, such as those described in (Polyak, 1964b; Zeiler, 2012; Duchi et al., 2011; Hinton et al., 2012; Kingma and Ba, 2015).

**Critical Gradient Stochastic Gradient Descent**   Critical Gradient Stochastic Gradient Descent ($\text{SGD}_\text{C}$) is the explicit integration of critical gradients into SGD. Specifically, the iteration comes as,

$$\theta_{t+1} = \theta_t - \alpha \cdot \texttt{aggr}(g_t, \, \mathbf{g_c}), \tag{1}$$

where $\mathbf{g_c}$ is the set of critical gradients and aggr denotes an aggregation function which is used to combine the critical gradients with the current-iteration gradient. We propose two possible functions for aggr including mean, the average of $g_t$ and all critical gradients, and sum, the addition of $g_t$ to the average of all critical gradients. Mathematically, for a buffer of size $C$ these are defined as

$$\texttt{mean}(g_t, \, \mathbf{g_c}) = \frac{1}{C+1}(g_t + \sum_{g_c \in \mathbf{g_c}} g_c) \quad (2) \qquad \texttt{sum}(g_t, \, \mathbf{g_c}) = g_t + \frac{1}{C} \sum_{g_c \in \mathbf{g_c}} g_c \quad (3)$$

In general, we find optimization to be robust to the specific choice of aggregation function, but we do observe that adaptive learning rate methods (Adam, RMSprop) perform best with mean, while convential SGD (with or without Polyak momentum) performs best with sum.

**Critical Gradients with Accelerated Methods**   Agarwal et al. (2019) show that curvature in adaptive gradients can be estimated without loss of generality with a finite moving window of past gradients. Following which we propose critical gradient extensions of SGD with Momentum (SGDM) (Eq.4), RMSprop (Eq.5), and Adam (Eq.6). The retention of the critical gradients can be naturally extended to more first-order optimization methods with the replacement of $g_t$ in those algorithms with aggr $(g_t, \, \boldsymbol{g_c})$. The update steps for the corresponding $_\text{C}$ versions are as follows:

$$\begin{aligned} m_t &= \gamma \cdot m_{t-1} + \texttt{aggr}(g_t, \boldsymbol{g_c}) \\ \theta_{t+1} &= \theta_t - \alpha \cdot m_t \end{aligned} \tag{4}$$

$$\begin{aligned} m_t &= \beta_1 m_{t-1} + (1 - \beta_1) \cdot \texttt{aggr}(g_t, \mathbf{g_c}) \\ v_t &= \beta_2 v_{t-1} + (1 - \beta_2) \cdot (\texttt{aggr}(g_t, \mathbf{g_c}))^2 \\ \hat{m}_t &= \frac{m_t}{1 - \beta_1^t} \\ \hat{v}_t &= \frac{v_t}{1 - \beta_2^t} \\ \theta_{t+1} &= \theta_t - \frac{\eta}{\sqrt{\hat{v}_t} + \epsilon} \hat{m}_t \end{aligned}$$

$$\begin{aligned} \mathbb{E}[g^2]_t &= 0.9 \cdot \mathbb{E}[g^2]_{t-1} + 0.1 \cdot \texttt{aggr}(g_t, \mathbf{g_c})^2 \\ \theta_{t+1} &= \theta_t - \frac{\eta}{\sqrt{\mathbb{E}[g^2]_t + \epsilon}} \cdot \texttt{aggr}(g_t, \mathbf{g_c}) \end{aligned} \tag{5}$$

$$\tag{6}$$

The general algorithm to integrate critical gradients into any optimizer is provided in Algorithm 1 in Appendix A. Replacing $g_t$ with aggr allows the optimization function to retain key gradients and optimize the parameters. The acceleration provided by the topC gradients complements the adaptive learning rate methods by not only computing the learning rate with respect to $g_t$ but also with respect to the critical gradients from the past. We observe that all $_C$-extended optimizers in our experiments in § 5 lead to faster convergence than their vanilla, non memory-augmented counterparts.

## 3   PROOF OF CONVERGENCE

Consider the $\text{SGD}_\text{C}$ update in Equation (1), where $\texttt{aggr}(g_t, \, \boldsymbol{g_c})$ outputs a linear combination of the critical gradients in memory $g_k \in \boldsymbol{g_c}$, and the stochastic gradient $g_t$, which is computed at the current parameters $\theta_t$. When the staleness of gradients in the memory is bounded, we can prove convergence of this method through the lens of *linear multi-step methods*. Formally, suppose there exists an integer

$K > 0$ such that $g_k \in \boldsymbol{g}_c$ at iteration $t$, implies $t - k < K$, and suppose the objective $F : \mathbb{R}^d \to \mathbb{R}$ is twice-continuously differentiable, $L$-smooth, and $\mu$-strongly convex, with $0 < \mu \leq L$. Assume that for all $k$, the stochastic gradient $g_k$ is a random vector satisfying

$$\mathbb{E}[g_k] = \nabla f(\theta_k) \quad \text{and} \quad \mathbb{E}\left[\|g_k - \nabla f(\theta_k)\|^2\right] \leq \sigma^2,$$

for some finite constant $\sigma^2$. Letting $\zeta_k := g_k - \nabla f(\theta_k)$ denote the gradient noise at iteration $k$, we take these gradient noise terms to be mutually independent. For examples of functions satisfying these properties, see, e.g., (Nesterov, 2004; Bubeck, 2015). Examples of typical tasks satisfying these assumptions are $\ell_2$-regularized logistic regression and $\ell_2$-regularized least-squares regression. Taken together, these properties imply that the Hessian $\nabla^2 F(\theta)$ exists, and for all $\theta \in \mathbb{R}^d$, the eigenvalues of $\nabla^2 F(\theta)$ lie in the interval $[\mu, L]$. In this case, let $\theta^\star$ denote the unique minimizer of $F$.

To prove convergence, we view $\text{SGD}_C$ as a linear multi-step method, i.e., the parameters at the current time step are updated with a linear combination of gradients from the past $K$ time steps. From this view, we can describe the evolution of the algorithm according to a discrete-time linear dynamical system, the convergence of which is characterized by the spectral properties of the system matrix. Our convergence theorem relies on the largest singular value of this system matrix, but can be strengthened to instead rely on the spectral radius of the system matrix by constructing an appropriate Lyapnuov function containing a time-invariant system matrix satisfying a particular Integral Quadratic Constraint (IQC) (Lessard et al., 2016). However, one downside of the IQC framework is that one must typically solve a semidefinite program to obtain explicit convergence rates (Hu and Lessard, 2017), whereas Theorem 1 (refer Appendix §B for proof) provides an analytical convergence rate.

**Theorem 1** (Linear Convergence of $\text{SGD}_C$). Let

$$q_{(\alpha,K)} := \sup_{\{w_k \in [0,1]\}, \{\eta_k \in [\mu,L]\}} \rho(\Lambda),$$

where $\rho(\Lambda)$ are the singular values of the numerical matrix

$$\Lambda := \begin{bmatrix} \lambda_0 & \lambda_1 & \cdots & \lambda_{K-1} \\ 0 & 1 & \cdots & 0 \\ \multicolumn{4}{c}{\dotfill} \\ 0 & 0 & \cdots & 1 \end{bmatrix} \in \mathbb{R}^{K \times K},$$

with $\lambda_0 = 1 - \alpha\eta_0$ and $\lambda_k = -\alpha w_k \eta_k$ for $k > 0$. If the step-size $\alpha > 0$ is chosen sufficiently small, such that $q_{(\alpha,K)} < 1$, then

$$\mathbb{E}\|\theta_{t+1} - \theta^\star\|^2 \leq q_{(\alpha,K)}^{2t} \|\theta_1 - \theta^\star\|^2 + \frac{\alpha^2 K}{1 - q_{(\alpha,K)}^2} \sigma^2,$$

where the expectation on the left hand side is with respect to the joint distribution over the random noise variables $\zeta_k, \ldots, \zeta_1$.

Theorem 1 shows that $\text{SGD}_C$ can be made to converge exponentially fast to a neighbourhood of the solution, the size of which is proportional to the variance of the stochastic gradients $\sigma^2$ and the step-size $\alpha^2$. If $K = 1$ (i.e., the memory $\boldsymbol{g}_c$, at all times $t$, only contains the mostly recently computed gradient), then $\text{SGD}_C$ reduces to regular stochastic gradient descent, and the rate $q_{(\alpha,K)}$ in Theorem 1 reduces to the well known convergence rate of SGD for smooth strongly convex functions $(\max_{\eta \in [\mu,L]} |1 - \alpha\eta L|)$, and the variance bound reduces to the standard variance bound for SGD with a constant step-size $(\alpha^2\sigma^2/(1 - q^2))$, see, e.g., (Bottou et al., 2018; Assran and Rabbat, 2020). For $K > 1$, it may be possible to choose the step-size $\alpha$ and the aggregation weights $\{w_k\}$ to obtain accelerated convergence rates, faster than SGD. For example, accelerated gradient methods such as Polyak and Nesterov momentum can be viewed as multi-step methods with $K = 2$. Note that although $K$ appears in the numerator of the coefficient of the variance term, it is also present in the rate term in the denominator, where faster convergence rates, smaller $q_{(\alpha,K)}$, directly lead to smaller variance bounds. Note that expressing the convergence rate of a multi-step method as the roots of a polynomial, as we do in Theorem 1, is not new; see, e.g., Polyak (1964a).

## 4 RELATED WORK

**Past-Gradients-Summarizing Optimizers** Several popular optimization algorithms incorporate the history of previous gradients through a summarizing variable. In practice, this is often done by

means of a decaying average, which is updated with each iteration. SGD with Momentum (SGDM) (Polyak, 1964b) uses this summary variable as a means of stabilizing descent along one path even if the current gradient points are uninformative. A variant of SGDM uses Nesterov momentum (NAG) (Nesterov, 1983; Sutskever et al., 2013) and applies the velocity to the current gradient as a correction factor. $\text{SGD}_\text{C}$ shares similarity with NAG except in $\text{SGD}_\text{C}$ the "momentum" is computed only using $C$ selected gradients. In a different use of a summarizing variable, AdaGrad (Duchi et al., 2011) keeps a running sum of *squared* gradients which is used to dynamically adjust the learning rate. AdaDelta (Zeiler, 2012) and RMSprop (Hinton et al., 2012) replace AdaGrad's running sum with a decaying average of the squared gradients. Adam (and AdaMax) (Kingma and Ba, 2015) uses a combination of momentum and a decaying-average-dependent learning rate. In general, the $_C$ variants ($\text{SGD}_\text{C}$, $\text{Adam}_\text{C}$, $\text{RMSprop}_\text{C}$, $\text{SGDM}_\text{C}$) provide an additional "layer" to the base algorithms that ensures any parameter computed within the algorithm incorporates the critical gradients. Although optimization using a fixed size buffer has been in existence, the novelty of the $_C$ variants comes from adding a heuristic to keep only the gradients that provide the largest displacements towards the minimum.

**Memory-Enhanced Optimizers**  Our algorithm belongs to a class of optimization methods which are aware of a limited history of gradients from previous iterations. One fellow algorithm from this class is Limited-History BFGS (LBFGS) (Nocedal, 1980), an optimization technique which approximates the Hessian of a function using a limited history of gradients, then uses the estimated Hessian to compute the descent direction for optimization. An online version of LBFGS (oLBFGS), well-suited for a machine learning context, was proposed by Schraudolph et al. (2007). These LBFGS methods utilize a moving window of past gradients, rather than our idea to maintain "critical" gradients based on a well-defined metric.

The storage of past gradients as a means to improve optimizer convergence is not a novel idea; both Stochastic Accelerated Gradient (SAG) (Roux et al., 2012) and SAGA (Defazio et al., 2014) do so in the context of accelerating stochastic gradient descent, while Stochastic Dual Coordinate Ascent (Shalev-Shwartz and Zhang, 2013) uses previous gradients for coordinate ascent. Empirically and theoretically, these techniques have been shown to yield good performance. The SAG and SAGA algorithms differ in their weight update step, but both involve storage of up to $n$ gradients, where $n$ is the number of training examples, which can be costly. Our method relies on the storage of a fixed number of gradients and uses a heuristic to decide whether to store a gradient in a dataset-size-independent memory. Our class of optimizers can natively run in a batch-based training loop unlike originally presented SAG and SAGA. Gazagnadou et al. (2019) address this issue in their recent work. We also extend our method beyond SGD to several other first-order optimization methods.

SAG and SAGA belong to the family of variance-reduced gradient-based algorithms (J. Reddi et al., 2015). Such algorithms theoretically and empirically outperform SGD, though with the trade-off of having a large cost in terms of memory or gradient evaluations. Some algorithms in this framework attempt to bridge both costs, for instance a variant of Stochastic Variance Reduced Gradient (SVRG) (Johnson and Zhang, 2013) stores intermediate gradients to avoid recomputing them later, and StochAstic Recursive grAdient algoritHm (SARAH) (Nguyen et al., 2017) uses a summarizing variable. Our method attempts to recapture the benefits of variance-reduced methods without significant computation or memory overhead.

**Optimizing Optimizers**  Our optimizer method also shares similarities with techniques which seek to optimize optimizers; that is, techniques which automatically learn the best way to accomplish the task of tuning the optimizer used in an outer-problem. The architecture proposed by Metz et al. (2020) uses an LSTM per tensor of the network paramters, with each LSTM being passed the gradient norms to influence the parameter update step, echoing our technique's use of gradient norm as a critical metric to parameter updates. In a similar vein, Andrychowicz et al. (2016) also use LSTMs, which are fed the complete gradient, and use the LSTM's recurrence as a means of implicitly accessing past gradients. Li and Malik (2017) uses a reinforcement learning-based approach and defines a state-space which includes a recent history of gradients. While our method equally maintains a ledger of past gradients, unlike this latter approach we use $\ell_2$-norm as a metric to pick out the critical gradients instead of arbitrarily keeping a recent history.

**Motivation for explicit memory**  Motivation for explicit memory-augmented optimizers over the ones that maintain an implicit summarization of history comes from memory-augmented neural

networks like Neural Turing Machines (NTMs) and their variants (Graves et al., 2014; 2016; Gulcehre et al., 2018). While a simple recurrent architecture like LSTM (Hochreiter and Schmidhuber, 1997) integrates information in its cell state which is a single vector, NTMs maintain a memory matrix which stores a set of such cell state vectors. This richer parameterization of memory helps NTMs in learning complex algorithmic tasks which are difficult to learn for an LSTM. Analogous to LSTMs and NTMs, SGDM and our $_C$-extended optimizers maintain a single vector and a set of vectors respectively as their memory. While our current formulation for memory-augmented optimizers uses heuristics to choose what information to store in the memory and how to use it, one could automatically learn both criteria and hence learn to optimize.

**Motivation for identifying critical gradients**    Likewise, recent work (Paul et al., 2021) has identified that samples with larger gradient norms can be used to identify a smaller set of training data that is important for generalization, and thereby prune the data as a pre-processing step. Our proposed approach of augmenting optimizers with memory buffer of critical gradients can be seen as a softer version of data-pruning in that the buffer retains gradients through the heuristic, thereby performing updates using only the important gradients throughout the entire training, without discarding data.

## 5  EXPERIMENTS

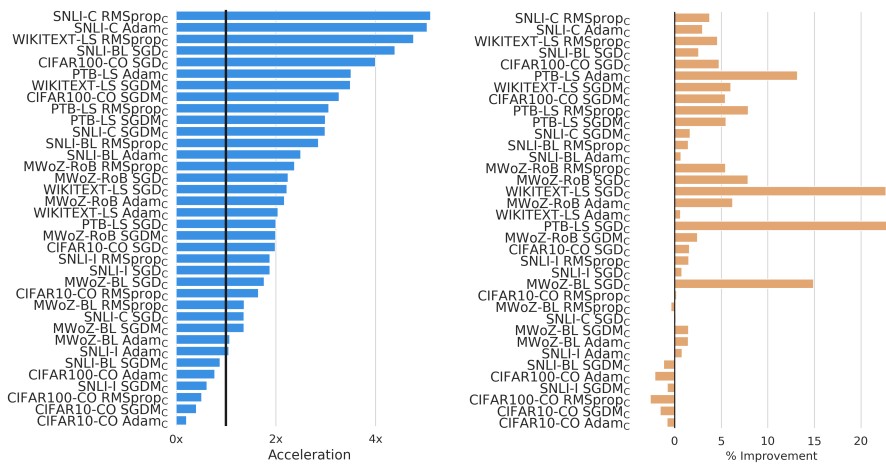

(a) Acceleration in Convergence        (b) Improvement over the Test performance

Figure 2: (a) Showcases the acceleration in convergence (the unit being the number of gradient updates with batch-size fixed) using a validation set provided by the $_C$ variants, compared pairwise (eg: Adam vs Adam$_C$), on the different tasks.(b) denotes the difference in performance of the models trained with $_C$ variants with respect to its corresponding base method on the test set. A positive value indicates that $_C$ variant performed better than its base method in that task. The side-by-side comparison allows us to observe that the proposed optimizers enabled accelerated convergence to better test results. Of the 9 tasks considered, the $_C$ optimizers performed the best in 6 tasks. Further, (b) shows that $_C$ version of the base optimizers enhanced the performance to as much as 23%.

We compare the proposed class of optimizers augmented with memory ($_C$ variants) with their vanilla versions on a variety of deep learning architectures. To understand the performance on common deep learning datasets, we experiment with shallow/deep convolutional neural network architectures (CO) on CIFAR 10/100 (Krizhevsky et al., 2009) respectively; Bi-LSTM (BL), InferSent (I), and a text-based convolutional architecture (C) on the Stanford Natural Language Inference (SNLI) (Bowman et al., 2015) dataset; LSTM on word level language modeling (LS) with the PennTreeBank (Marcus et al., 1993), and WikiText (Merity et al., 2016) datasets; RoBERTa-Base (RoB) and Bi-LSTM (BL) on language generation in dialogue task with MultiWoZ 2.0 dataset (Budzianowski et al., 2018). Additionally, we preform analysis experiments using logistic regression (LR) and multi layer perceptrons (MLP) on the MNIST digit classification dataset (LeCun and Cortes, 2010). Across the experiments, we compare all the 8 optimizers averaged over 5 different runs with different

seed values— Adam, SGD, SGDM, RMSprop, $\text{Adam}_C$, $\text{SGD}_C$, $\text{SGDM}_C$, and $\text{RMSprop}_C$ on 9 tasks — CIFAR100, CIFAR10, SNLI-I, SNLI-C, SNLI-BL, MWoZ-RoB, MWoZ-BL, PTB-LS, and WIKITEXT-LS*. For a fair comparison, we follow the standard experimental protocol for all the experiments and tune all the hyperparameters of an optimizer individually for every task.

Our experimental results are aggregated from 5 independent runs, with the hyperparameters for each optimizer extensively tuned. This involves tuning the learning rate in all optimizers, the `topC` and `decay` parameters in all $_C$ algorithms, and all optimizer-specific hyperparameters in both the vanilla versions and their $_C$ counterparts. A full description of the values used to tune the various parameters, architecture and dataset details are in Appendix §E, §D, and §C respectively. The best set of hyperparameters was selected based on which ones yielded the best validation performance (i.e. highest accuracy, highest BLEU score, or lowest perplexity).

Figure 2(a) shows that the proposed $_C$ variants provided a consistent acceleration in convergence over their vanilla versions. The accelerated convergence also resulted in improved test performance, as highlighted in Figure 2(b) (Exhaustive list is provided in Appendix §G). In addition to the aforementioned improvements in vanilla-vs-$_C$ comparisons, the $_C$ optimizers yielded the best test performance across all 8 optimizers in 6/9 tasks. Of the three remaining tasks, the $_C$ versions performed close to the vanilla versions. In the case of $\text{MWoZ} - \text{BL RMSprop}_C$ the $_C$ version showed marginally accelerated convergence close to the optimal solution. Furthermore, Figure 3 compares the validation performance after each epoch when trained with the different optimizers on the different tasks. One immediate observation was that in language experiments – SNLI, PTB, MWoZ – the $_C$ versions stayed above most if not all of the vanilla optimizers. While that separation is not as clear in the vision tasks, we observed that the pairwise comparison mostly put the $_C$ optimizer at an advantage in accelerated convergence.

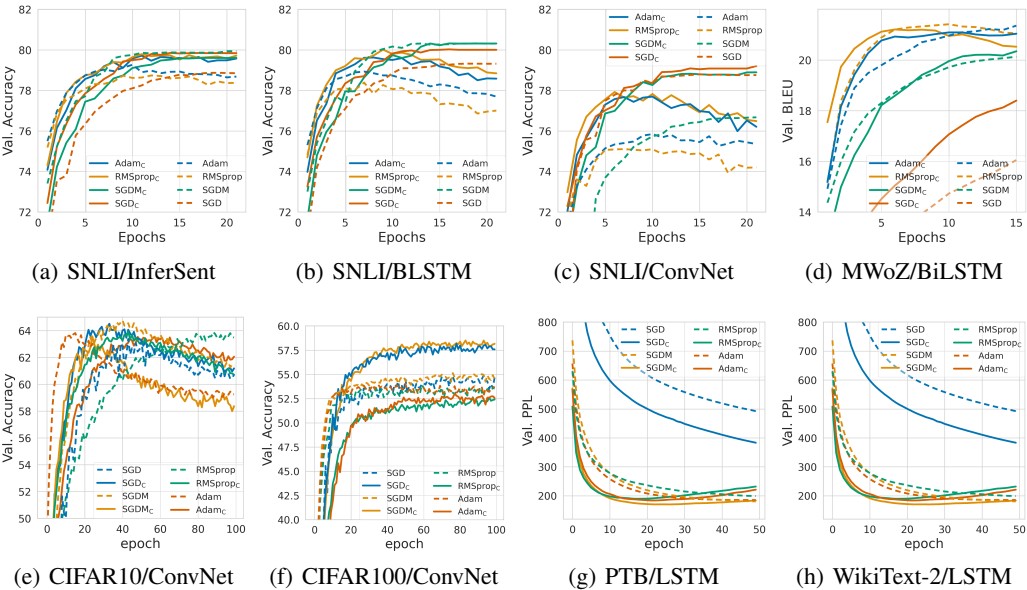

(a) SNLI/InferSent    (b) SNLI/BLSTM    (c) SNLI/ConvNet    (d) MWoZ/BiLSTM

(e) CIFAR10/ConvNet    (f) CIFAR100/ConvNet    (g) PTB/LSTM    (h) WikiText-2/LSTM

Figure 3: The validation performance comparison of the proposed optimizers and their vanilla versions showed that, in general, the proposed optimizers showed accelerated convergence than their counterparts. Also, in the 3 tasks — SNLI-I (a), SNLI-BL (b) and MWoZ-BL (d) where the $_C$ methods are not the optimal — only falls short by a negligible margin. For ease of visualization, confidence intervals across seeds are suppressed.

---

*Code to reproduce results: https://github.com/chandar-lab/CriticalGradientOptimization.
Optimizer package: https://github.com/chandar-lab/CGOptimizer

# 6 ANALYSIS

**Buffer Staleness Bound** A key assumption behind the theoretical convergence of $\mathrm{SGD_C}$ is the existence of an upper bound on the buffer gradients' staleness (i.e the number of iterations a gradient remains in the critical buffer). We analysed the distribution of average number of steps a gradient stays in the memory for the different values of `topC` and `decay` in Figure 4. We observe that tuning the parameters ensures that gradients in the buffer are refreshed frequently, allowing for a renewal of information. Later, in analysing the effect of `decay` and `topC` on the performance in a task, we observe that the higher performance of the $_C$ methods correlate with the findings of this analysis.

**Use of Critical Gradients** Our Critical Gradient method utilizes a buffer which is filled by using a norm-based, "King of the Hill" sampling of incoming gradients, i.e. gradients always stored and sorted in decreasing order by $\ell_2$-norm, with the smallest norm always being the one removed. We conduct ablation studies to probe the three assumptions of critical gradients using LSTM architecture with PTB dataset.: (a) the use of norm as a controlling metric, (b) the removal of the smallest-norm entry when adding to a full buffer, and (c) the update step depending on an ensemble of buffer gradients via the mean rather than on select gradients.

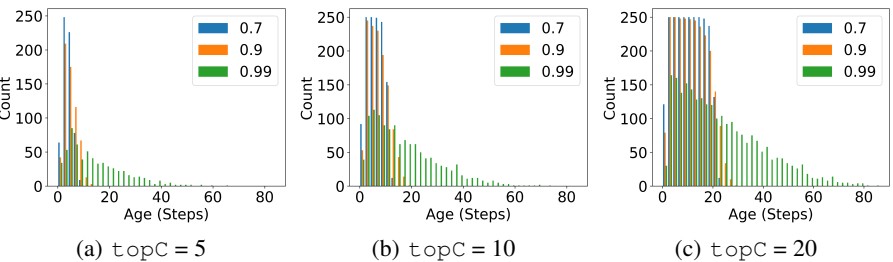

Figure 4: Histograms showing ages of the buffer as recorded at the end of every epoch, across five seeds for several `decay` values, for a MLP on MNIST. $0.99$ decays at a much slower rate, keeping stale gradients in the memory, while $0.7$ replaces the gradients frequently. Also, the size of the buffer (`topC` ), as expected, correlates directly with retention of stale gradients.

Addressing assumption (a) we test the use of maximal norm sampling (Figure 5(a)) by comparing with Cosine Similarity Sampling (CSS), Cosine Diversity Sampling (CDS), Mean-Norm Diversity Sampling (MNDS), and random sampling via "coin toss". Figure 5(b) depicts our algorithm alongside different gradient replacement methods, including Random Replacement (RR) and Norm-Controlled Probabilistic Replacement (NCPR), as well as comparing against keeping a running First-In-Fist-Out (FIFO) queue and only using the smallest ("`bottomC`") gradients. Finally we compare the typical optimizer aggregation method with the buffer gradients minimum, maximum, and median norm (Figure 5(c)). A full description of these techniques can be found in Appendix §F.9.

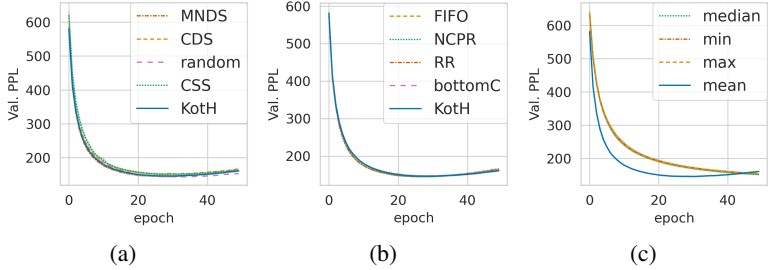

Figure 5: (a) compares gradient selection techniques based on different metrics, (b) compares different replacement strategies when flushing out a gradient from memory and (c) compares choice of aggregating the gradients in the buffer. The algorithm converges in all the different choices.

We observe in most cases, the sampling (a) and replacement (b) strategies for gradients in the buffer and all ablations converge regardless of these considerations. While the Critical Gradients formulation is not unique as a high-performing implementation of a memory-augmented optimizer, it remains conceptually straightforward, easy to implement, and the buffer properties are deterministic as compared to some of our ablation studies which were probabilistic. Although, a rigorous analysis is necessary to verify the generality of this result.

Experiments comparing the performance of mean-based aggregation against aggregation based on individual gradients show that aggregation based on the ensemble must be employed, and that the optimizers benefit from the added information.

**Effect of hyperparameters**  We study the performance's sensitivity to `decay` and `topC` using $\text{Adam}_C$ on different tasks.

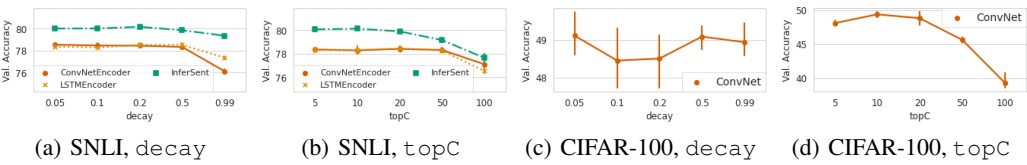

| (a) SNLI, `decay` | (b) SNLI, `topC` | (c) CIFAR-100, `decay` | (d) CIFAR-100, `topC` |

Figure 6: We observe that non-trivial choices of `decay` and `topC` provided best performance for $\text{Adam}_C$ with different architectures on SNLI task. Also, the usefulness of explicit memory aligns with the evidence on gradient staleness from Figure 4.

The `decay` $\in [0, 1)$ parameter allows to tune the retention of critical gradient information from the past. Setting `decay` $= 0$ reduces the priority of new gradients added to the buffer, making the $_C$ variants behave like an average over a moving window of length `topC`. On the other extreme, `decay` $= 1$ does not decrease the priorities of the past gradients and allows them to be in the buffer for a much longer period hindering the convergence of the optimizers. Also, `decay` maintains the staleness of the gradients by exponentially decaying their priority in the memory buffer. We experimented with $\text{Adam}_C$ optimizer for the different values of `decay` with every other parameter fixed (Figure 6). We observed that `decay` values closer to 1 did not have any advantage at best or leads to inferior performance on an unseen set at worst in most of the tasks.

`topC` $\in [0, \infty)$ defines the size of the buffer that holds the past gradients. When `topC` is set to 0, $_C$ variants behave like the base algorithms and when `topC` is the size of the dataset, the optimizer becomes SAG-like. Also, `topC` is directly proportional to the staleness of the gradients in the buffer. A higher `topC`, though, guarantees better expressibility also becomes responsible for more stale gradients in the buffer. Bengio et al. (2020) also observes a similar issue in momentum used in TD-learning and proposes a correction factor to alleviate the issue. Here, the staleness is contained with the parameters `topC` and `decay`. As an empirical evidence, we observe that the optimizers perform well on the validation set with lower `topC` values. In most of the experiments, `topC` $> 20$ was either not useful or was hurting the performance (also in Appendix F).

## 7  CONCLUSION

We propose a general method to enhance the performance of an optimizer with an augmented memory buffer to maintain a limited set of critical gradients from the history. The proposed memory mechanism, when integrated with several state-of-the-art optimizers, accelerated the learning in all cases and also improved their performance in several cases. The $_C$ family of optimizers proposed in this paper are the first steps towards designing smart optimizers that can learn to optimize. Some of the immediate future research directions include learning *what* to store in the memory and learning *how* to integrate the information in memory to the gradient update steps.

**Acknowledgements**  We would like to acknowledge Compute Canada and Calcul Quebec for providing computing resources used in this work. The authors would also like to thank members of Chandar Research Lab, Mila for helping with the code reviews and reviewing the manuscripts. SC is supported by a Canada CIFAR AI Chair and an NSERC Discovery Grant.

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

## A  GENERAL CRITICAL GRADIENT ALGORITHM

We present the pseudocode for the Critical Gradient algorithm below.

---

**Algorithm 1** Critical Gradients Optimization

---

**Define** #epochs $E$, #gradient updates $B$, model $\theta$, learning rate $\alpha$
**Define** optimizer $O(g_t, \theta_t, \mathbf{h})$, which produces a parameter update given gradient $g_t$, parameter $\theta_t$ and hyperparameters $\mathbf{h}$
**Define** aggregation function $\text{aggr}(g_t, \mathbf{g_c})$ which outputs a linear combination of $g_t$ and the entries in $\mathbf{g_c}$
**Initialize** empty critical gradient buffer $\{\mathbf{g_c}\}$ with decay rate $d$
**for** $s = 0$ **to** $E$ **do**
   **for** $t = 0$ **to** $B$ **do**
      **Sample** an input-output pair $(x_i, y_i)$
      $g_t \leftarrow \nabla_\theta L_i(\theta_t)$
      $\theta_{t+1} \leftarrow O(\text{aggr}(g_t, \mathbf{g_c}), \theta_t, \mathbf{h})$
      **if** $\{\mathbf{g_c}\}$ is not full **then**
         Add $g_t$ to $\{\mathbf{g_c}\}$
      **else**
         **if** $\|g_t\|_2 > \|\min(\{\mathbf{g_c}\})\|_2$ **then**
            Remove $\min(\{\mathbf{g_c}\})$ from $\{\mathbf{g_c}\}$
            Add $g_t$ to $\{\mathbf{g_c}\}$
         **end if**
      **end if**
      **for** $g_c \in \{\mathbf{g_c}\}$ **do**
         $\|g_c\|_2 \leftarrow d \cdot \|g_c\|_2$
      **end for**
   **end for**
**end for**

---

## B  CONVERGENCE PROOF

*Proof.* Recall that if $F : \mathbb{R}^d \to R$ is twice continuously differentiable, then for all $x, y \in \mathbb{R}^d$

$$\nabla F(y) = \nabla F(x) + \int_0^1 \nabla^2 F(x + t(y - x)) \mathrm{d}t \ (y - x). \tag{7}$$

Letting $r_k := \theta_k - \theta^\star$ denote the suboptimality of the parameters at iteration $k$ and applying (7) with $y = \theta_k$ and $x = \theta^\star$, we get that

$$g_k = H_k r_k + \zeta_k, \quad \text{where} \quad H_k = \int_0^1 \nabla^2 f(\theta^\star + t r_k) \mathrm{d}t,$$

where $\zeta_k$ denotes the gradient noise at iteration $k$. Using this and bounded staleness assumption, we have that

$$\text{aggr}(g_t, \mathcal{G}_t) = \sum_{k=0}^{K-1} w_{t,k} g_{t-k} = \sum_{k=0}^{K-1} w_{t,k}(H_{t-k} r_{t-k} + \zeta_{t-k}), \tag{8}$$

where $\{w_{t,k}\}_{k=0}^{K-1}$ are non-negative scalars in the closed interval $[0, 1]$ used to linearly aggregate the gradients in memory with the most recent gradient $g_t$. For gradients $g_{t-k}$ that are not stored in memory, the corresponding aggregation scalar $w_{t,k}$ is equal to 0. Without loss of generality, we take $w_{t,0} = 1$, since we can always incorporate it into the step-size $\alpha$ and re-scale the other weights $w_{t,1}, \ldots, w_{t,K-1}$ accordingly. Substituting (8) into (1) and subtracting $\theta^\star$ from each side, we get that

$$r_{t+1} = r_t - \alpha \sum_{k=0}^{K-1} w_{t,k}(H_{t-k} r_{t-k} + \zeta_{t-k}).$$

Thus, we have that $\text{SGD}_\text{C}$ evolves according to the linear system

$$\begin{bmatrix} r_{t+1} \\ r_t \\ \vdots \\ r_{t+1-K} \end{bmatrix} = A_t \begin{bmatrix} r_t \\ r_{t-1} \\ \vdots \\ r_{t-K} \end{bmatrix} - \alpha \begin{bmatrix} \sum_{k=0}^{K-1} w_{t,k} \zeta_{t-k} \\ 0 \\ \vdots \\ 0 \end{bmatrix} \quad \text{where} \quad A_t = \begin{bmatrix} T_{t,0} & T_{t,1} & \cdots & T_{t,K-1} \\ 0 & I & \cdots & 0 \\ \cdots\cdots\cdots\cdots\cdots\cdots\cdots\cdots \\ 0 & 0 & \cdots & I \end{bmatrix},$$

with $T_{t,0} = I - \alpha H_t$ and $T_{t,k} = -\alpha w_{t,k} H_{t-k}$ for $k = 1, \ldots, K - 1$. Unrolling the recursion, we have that

$$
\begin{bmatrix} r_{t+1} \\ r_t \\ \vdots \\ r_{t+1-K} \end{bmatrix} = (A_t \cdots A_1) \begin{bmatrix} \theta_1 - \theta^\star \\ 0 \\ \vdots \\ 0 \end{bmatrix} - \alpha \begin{bmatrix} \sum_{k=0}^{K-1} w_{t,k} \zeta_{t-k} \\ 0 \\ \vdots \\ 0 \end{bmatrix} - \alpha \sum_{j=1}^{t-1} (A_t \cdots A_{j+1}) \begin{bmatrix} \sum_{k=0}^{K-1} w_{j,k} \zeta_{j-k} \\ 0 \\ \vdots \\ 0 \end{bmatrix}, \tag{9}
$$

from which it is clear that we may expect convergence properties to depend on the spectral properties of the matrices $\{A_k\}_{k=1}^t$. The convergence rate of $\text{SGD}_\text{C}$ will therefore depend on the largest singular value of the matrices $\{A_k\}_{k=1}^t$. From Polyak (1964a, Lemma 4), we have that

$$
\|A_j\| \leq \sup_{\{w_k \in [0,1]\}, \{\eta_k \in [\mu, L]\}} \rho(\Lambda), \tag{10}
$$

where $\rho(\Lambda)$ are the singular values of the numerical matrix

$$
\Lambda := \begin{bmatrix} \lambda_0 & \lambda_1 & \cdots & \lambda_{K-1} \\ 0 & 1 & \cdots & 0 \\ \cdots & \cdots & \cdots & \cdots \\ 0 & 0 & \cdots & 1 \end{bmatrix} \in \mathbb{R}^{K \times K},
$$

with $\lambda_0 = 1 - \alpha \eta_0$ and $\lambda_k = -\alpha w_k \eta_k$ for $k = 1, \ldots, K - 1$. Note that Polyak (1964a, Lemma 4) originally refers to the eigenvalues of a block matrix, however, in the case where the blocks are commutative, it is straightforward to extend the lemma to describe the singular values. To see this, simply apply the original lemma to the eigenvalues of the matrix $\Lambda^\top \Lambda$, where commutativity of the individual blocks ($T_{j,0}, \ldots, T_{j,K}$ are all symmetric and therefore commutative) allows you to specify the eigenvalues of the new block matrix in terms of the eigenvalues of the original blocks.

Letting $q_{(\alpha, K)}$ denote the right hand side of (10), we have that $\|A_j\| \leq q_{(\alpha, K)}$ for all $j = 1, \ldots, t$, and $q_{(\alpha, K)}$ has an analytic definition as the roots of a polynomial. Taking 2-norms, applying submultiplicativity of matrix norms, and using the bounded variance assumption along with the fact that the noise terms $\{\zeta_k\}_{k=0}^t$ are mutually independent and that the aggregation weights $\{w_{t,k}\}_{t \geq 0, k \in [1,K)}$ are in the closed interval $[0, 1]$ for all $t, k$, we have that

$$
\mathbb{E} \|r_{t+1}\|^2 \leq q_{(\alpha, K)}^{2t} \|\theta_1 - \theta^\star\|^2 + \alpha^2 \sigma^2 K \sum_{j=1}^t q_{(\alpha, K)}^{2(t-j)}, \tag{11}
$$

where the expectation on the left hand side is with respect to the joint distribution over the random noise variables $\zeta_k, \ldots, \zeta_1$.

By assumption, since the step-size $\alpha$ is chosen such that $q_{(\alpha, K)} < 1$, equation (11) simplifies as

$$
\mathbb{E} \|r_{t+1}\|^2 \leq q_{(\alpha, K)}^{2t} \|\theta_1 - \theta^\star\|^2 + \frac{\alpha^2 K}{1 - q_{(\alpha, K)}^2} \sigma^2,
$$

where we have implicitly used the upper bound on the limit of a geometric sequence. $\blacksquare$

## C  DATA SET DISTRIBUTIONS

Table 1 details the train-valid-test splits of all the data sets used in the experiments.

| Dataset | #Train | #Valid | #Test |
|---|---|---|---|
| covtype | 5000 | N/A | N/A |
| rcv1 | 5000 | N/A | N/A |
| MNIST | 50K | 10K | 10K |
| CIFAR-10 | 40K | 10K | 10K |
| CIFAR-100 | 40K | 10K | 10K |
| SNLI | 550K | 10K | 10K |
| WikiText | 2M | 213K | 241K |
| Penn TreeBank | 890K | 70K | 78K |
| MultiWoZ | 115K (1.5M) | 20K (200K) | 20K (200K) |

Table 1: The splits of the data sets in number of samples. For WikiText, Penn TreeBank and MultiWoZ as the models are trained on a language modeling objective, the splits are given in number of tokens. For MultiWoZ, the number of utterances is provided with the number of tokens in parentheses.

# D  EXPERIMENT DETAILS

We tested the proposed class of optimizers on different tasks that have potentially different loss surfaces in large parameter spaces. To cast a wide net and ensure that we capture a plethora of neural architectures, we vary the network designs by using fully-connected, recurrent (Rumelhart et al., 1985), convolutional (LeCun et al., 1999), dropout (Srivastava et al., 2014), ReLU (Agarap, 2018) layers and a large Transformer language model architecture – RoBERTa-Base (Liu et al., 2019) across different tasks. A brief description of the data and architectures used follows.

CIFAR-10 is an image classification task with 10 classes. We train a shallow Convolutional Neural Network (ConvNet) with dropout and ReLU activation using the different optimizers and their $_C$ variants.

CIFAR-100 is an image dataset similar in size to CIFAR-10 but with 100 classes of images. This task was approached using a Convolutional Neural Network with three batch-normalized convolutional blocks and one fully-connected block with dropout applied between blocks.

We experiment with the commonly-used language inference dataset, Stanford Natural Language Inference (SNLI) dataset, to compare the performances of $_C$ variants training on three different text encoder architectures: 4-layer convolutional network (ConvNetEncoder), 2 layer bi-directional encoder with 2 linear projection layers (InferSent) and a unidirectional LSTM with 2 recurrent layers (LSTMEncoder). The classifier on top of the representations learned by the encoder architectures is a 2-layer fully-connected Multi-Layer Perceptron with dropout connection.

The Penn Tree Bank (PTB) is a syntax-annotated text corpus sourced from stories from the Wall Street Journal. We use this corpus for a word-level language modeling task using a 1-layer LSTM with dropout. Gradient clipping is employed to avoid exploding gradients. We evaluate the model by measuring perplexity (PPL) in the validation set; lower PPL scores are preferred.

MultiWoZ 2.0 is a popular dataset that has human-to-human goal-oriented conversations on different topics. The objective is to generate the next utterance conditioned on the history of utterances in the conversation. We experiment with a very large language model architecture – RoBERTa-Base – and a BiLSTM Sequence-to-Sequence architecture (Vinyals and Le, 2015) for the next utterance prediction. RoBERTa was trained with CausalLM-Head while BiLSTM model was an encoder-decoder architecture with a Bi-LSTM encoder and LSTM with Attention decoder.

We use models with varying size of trainable parameters as shown in Table 2.

Table 2: We experimented with models with different sizes for a comprehensive study of the proposed $_C$ variants.

| Model | #Params | Dataset(s) |
|---|---|---|
| Logistic Regression | 8K (55/64) | MNIST (rcv1/covtype) |
| NeuralNetwork | 25K | MNIST |
| ConvNet | 600K (62K) | CIFAR-100 (CIFAR-10) |
| LSTM | 20K (600K) | PTB (WikiText) |
| LSTMEncoder | 600K | SNLI |
| InferSent | 1.2M | SNLI |
| ConvNetEncoder | 3.1M | SNLI |
| RoBERTa-Base | 125M | MultiWoZ |
| Bi-LSTM | 574K | MultiWoZ |

We use the same hyperparameter initialization for comparisons with base optimization methods and tune the learning rate hyperparameter using grid search. The primary objective of the experiments is to verify the consistency of convergence to better solutions across tasks and architectures. All reported results are averaged over 5 different runs with different random seed values.

## D.1  MODEL HPARAMS

All experiments are averaged for 5 different runs with different seeds. We use PyTorch 1.1 for the experiments and use their implementation of the base optimizers available in `torch.optim`. The details of hyperparameters used for the model are in Table 3.

Table 3: Architecture details for models used in experiments.

| Model | Dataset | #Layers | #Hidden | ReLU/Dropout |
|---|---|---|---|---|
| Log. Reg. | covtype | 1 | N/A | No/No |
| Log. Reg. | rc1 | 1 | N/A | No/No |
| Log. Reg. | MNIST | 1 | N/A | No/No |
| Log. Reg. | synthetic | 1 | N/A | No/No |
| MLP | MNIST | 2 | 32 | Yes/No |
| CNN | CIFAR-10 | 5 | 120 | Yes/Yes |
| CNN | CIFAR-100 | 9 | 4096 | Yes/Yes |
| LSTM | PTB | 1 | 128 | No/No |
| LSTM | WikiText | 1 | 128 | No/No |
| ConvNetEnc. | SNLI | 2 | 200 | Yes/Yes |
| LSTMEncoder | SNLI | 2 | 200 | No/Yes |
| InferSent | SNLI | 2 | 200 | Yes/Yes |
| RoBERTa-Base | MultiWoZ | 12 | 768 | Yes/Yes |
| Bi-LSTM Attn | MultiWoZ | 4 | 200 | No/No |

## D.2 RUNTIME STATISTICS

We logged the approximate time for each epoch for different values of `topC` across the different models. Although the results are populated from the experiments with Adam and its $C$ variants, the results can be extended to the other optimizers and its variants. These times are reported in Table 4.

Table 4: Approximate time taken for one epoch of training of the models in the public repositories on the different data sets. The time is clocked in minutes. The time per epoch increases linearly with increase in `topC` over the (**B**)ase method. Although for lower values of `topC` the time per epoch is smaller, there is still room for improvement to bring down the time with smart update rules as discussed in §7.

| Dataset | Model | B | C5 | C10 | C20 | C20 | C100 |
|---|---|---|---|---|---|---|---|
| MNIST | Log.Reg | 0.3 | 0.4 | 0.5 | 0.6 | 0.9 | .5 |
| | Neural-Net | 0.3 | 0.5 | 0.6 | 0.8 | 1.4 | 2.3 |
| PTB | LSTM | 0.08 | 0.4 | 0.6 | 0.9 | 2 | 6 |
| WikiText | LSTM | 0.6 | 1 | 1.4 | 2.6 | 5 | 10 |
| CIFAR-10 | CNN | 0.5 | 0.9 | 1 | 1.5 | 3 | 5 |
| CIFAR-100 | CNN | 0.75 | 1.5 | 2 | 4 | 6 | 12 |
| | LSTMEnc. | 1 | 3 | 6 | 10 | 25 | 45 |
| SNLI | InferSent | 2 | 4 | 9 | 20 | 35 | 50 |
| | ConvNet | 1 | 4 | 8 | 20 | 40 | 65 |
| MultiWoZ | RoBERTa | 15 | 25 | 35 | 50 | NA | NA |
| | BiLSTM | 3 | 4 | 5 | 5 | 7 | 10 |

# E HYPERPARAMETERS

## E.1 RANGE OF HPARAMS

We present the range of hyperparameters used in our experiments (Table 5). Optimizer-specific parameters were used on both their vanilla and $C$ versions.

Table 5: The hyperparameter configurations of the different optimizers in the experiments.

| Hparam | Choices |
|---|---|
| **Adam** $\beta_1$ | $\{0.9, 0.99, 0.999\}$ |
| **Adam** $\beta_2$ | $\{0.99, 0.999, 0.9999\}$ |
| **RMSprop** $\alpha$ | $\{0.9, 0.99, 0.999\}$ |
| **SGDM** $momentum$ | $\{0.9, 0.99, 0.999\}$ |
| $learning\_rate$ | $\{0.1, 0.01, 0.001, 0.0001, 0.00001\}$ |
| decay | $\{0.7, 0.9, 0.99\}$ |
| topC | $\{5, 10, 20\}$ |

## E.2 HPARAMS OF THE BEST CONFIGURATIONS

The learning rates and other hyperparameters of the optimizers used in the results reported in the paper are listed in Tables 6, 7, 8 and 9.

Table 6: Best hyperparameter configurations for CIFAR-10 and CIFAR-100 image classification tasks.

| Dataset | Model | Optimizer | Learning Rate | topC | decay | Momentum | $\beta_1$ | $\beta_2$ | $\alpha$ |
|---|---|---|---|---|---|---|---|---|---|
| CIFAR-10 | Shallow ConvNet | $Adam_C$ | 0.0001 | 5 | 0.7 | N/A | 0.99 | 0.99 | N/A |
| | | Adam | 0.001 | N/A | N/A | N/A | 0.9 | 0.99 | N/A |
| | | $RMSprop_C$ | 0.0001 | 5 | 0.7 | N/A | N/A | N/A | 0.9 |
| | | RMSprop | 0.0001 | N/A | 0 | N/A | N/A | N/A | 0.9 |
| | | $SGDM_C$ | 0.001 | 5 | 0.7 | 0.9 | N/A | N/A | N/A |
| | | SGDM | 0.001 | N/A | N/A | 0.9 | N/A | N/A | N/A |
| | | $SGD_C$ | 0.01 | 5 | 0.99 | N/A | N/A | N/A | N/A |
| | | SGD | 0.01 | N/A | N/A | N/A | N/A | N/A | N/A |
| CIFAR-100 | Deep ConvNet | $Adam_C$ | 0.00001 | 20 | 0.7 | N/A | 0.9 | 0.9999 | N/A |
| | | Adam | 0.0001 | N/A | N/A | N/A | 0.9 | 0.9999 | N/A |
| | | $RMSprop_C$ | 0.00001 | 20 | 0.7 | N/A | N/A | N/A | 0.99 |
| | | RMSprop | 0.0001 | N/A | N/A | N/A | N/A | N/A | 0.99 |
| | | $SGDM_C$ | 0.001 | 20 | 0.7 | 0.9 | N/A | N/A | N/A |
| | | SGDM | 0.01 | N/A | N/A | 0.9 | N/A | N/A | N/A |
| | | $SGD_C$ | 0.01 | 5 | 0.9 | N/A | N/A | N/A | N/A |
| | | SGD | 0.1 | N/A | N/A | N/A | N/A | N/A | N/A |

Table 7: Best hyperparameter configurations for language modelling tasks on WikiText and PennTree-Bank datasets.

| Dataset | Model | Optimizer | LR | topC | decay | Momentum | $\beta_1$ | $\beta_2$ | $\alpha$ |
|---|---|---|---|---|---|---|---|---|---|
| PTB | LSTM | $Adam_C$ | 0.0001 | 5 | 0.9 | N/A | 0.9 | 0.999 | N/A |
| | | Adam | 0.0001 | 0 | 0 | N/A | 0.9 | 0.999 | N/A |
| | | $RMSprop_C$ | 0.0001 | 20 | 0.9 | N/A | N/A | N/A | 0.9 |
| | | RMSprop | 0.0001 | 0 | 0 | N/A | N/A | N/A | 0.9 |
| | | $SGDM_C$ | 0.1 | 10 | 0.9 | 0.9 | N/A | N/A | N/A |
| | | SGDM | 0.1 | 0 | 0 | 0.9 | N/A | N/A | N/A |
| | | $SGDM_C$ | 0.1 | 2 | 0.95 | N/A | N/A | N/A | N/A |
| | | SGD | 0.1 | 0 | 0 | N/A | N/A | N/A | N/A |
| Wikitext | | $Adam_C$ | 0.0001 | 20 | 0.9 | N/A | 0.9 | 0.9999 | N/A |
| | | Adam | 0.0001 | 0 | 0 | N/A | 0.9 | 0.9999 | N/A |
| | | $RMSprop_C$ | 0.0001 | 20 | 0.9 | N/A | N/A | N/A | 0.9 |
| | | RMSprop | 0.0001 | 0 | 0 | N/A | N/A | N/A | 0.9 |
| | | $SGDM_C$ | 0.1 | 10 | 0.9 | 0.9 | N/A | N/A | N/A |
| | | SGDM | 0.1 | 0 | 0 | 0.9 | N/A | N/A | N/A |
| | | $SGDM_C$ | 0.1 | 2 | 0.95 | N/A | N/A | N/A | N/A |
| | | SGD | 0.1 | 0 | 0 | N/A | N/A | N/A | N/A |

Table 8: Best hyperparameter configurations for text generation for dialogue on MultiWoZ2.0 dataset.

| Dataset | Model | Optimizer | LR | topC | decay | Momentum | $\beta_1$ | $\beta_2$ | $\alpha$ |
|---|---|---|---|---|---|---|---|---|---|
| MultiWoZ2.0 | RoBERTa-Base | $Adam_C$ | 0.00001 | 5 | 0.7 | N/A | 0.99 | 0.999 | N/A |
| | | Adam | 0.00001 | 0 | 0 | N/A | 0.99 | 0.999 | N/A |
| | | $RMSprop_C$ | 0.0001 | 5 | 0.7 | N/A | N/A | N/A | 0.99 |
| | | RMSprop | 0.00001 | 0 | 0 | N/A | N/A | N/A | 0.99 |
| | | $SGDM_C$ | 0.01 | 5 | 0.7 | 0.9 | N/A | N/A | N/A |
| | | SGDM | 0.01 | 0 | 0 | 0.9 | N/A | N/A | N/A |
| | | $SGD_C$ | 0.1 | 5 | 0.7 | N/A | N/A | N/A | N/A |
| | | SGD | 0.1 | 0 | 0 | N/A | N/A | N/A | N/A |
| | Bi-LSTM | $Adam_C$ | 0.001 | 5 | 0.7 | N/A | 0.99 | 0.999 | N/A |
| | | Adam | 0.001 | 0 | 0 | N/A | 0.99 | 0.999 | N/A |
| | | $RMSprop_C$ | 0.001 | 5 | 0.7 | N/A | N/A | N/A | 0.99 |
| | | RMSprop | 0.001 | 0 | 0 | N/A | N/A | N/A | 0.99 |
| | | $SGDM_C$ | 0.1 | 5 | 0.7 | 0.9 | N/A | N/A | N/A |
| | | SGDM | 0.1 | 0 | 0 | 0.9 | N/A | N/A | N/A |
| | | $SGD_C$ | 0.1 | 5 | 0.7 | N/A | N/A | N/A | N/A |
| | | SGD | 0.1 | 0 | 0 | N/A | N/A | N/A | N/A |

# F ANALYSIS

## F.1 NOTE ON AGGREGATION

The aggregation function `aggr` lies at the core of integrating critical gradients into an algorithm. In §2 we presented `mean` and `sum` function for aggregation. The notable difference in these methods is the importance of the current gradient, as it is more heavily weighted with `sum` (which does not scale it) than with `mean` (which scales it by a factor of $\frac{1}{topC+1}$. While our algorithms converged

Table 9: Best hyperparameter configurations for text classification for language inference on SNLI dataset.

| Dataset | Model | Optimizer | LR | topC | decay | Momentum | $\beta_1$ | $\beta_2$ | $\alpha$ |
|---------|-------|-----------|----|------|-------|----------|-----------|-----------|----------|
| SNLI | InferSent | $\text{Adam}_C$ | 0.0001 | 5 | 0.7 | N/A | 0.99 | 0.999 | N/A |
| | | Adam | 0.001 | 0 | 0 | N/A | 0.99 | 0.999 | N/A |
| | | $\text{RMSprop}_C$ | 0.0001 | 5 | 0.7 | N/A | N/A | N/A | 0.99 |
| | | RMSprop | 0.001 | 0 | 0 | N/A | N/A | N/A | 0.99 |
| | | $\text{SGDM}_C$ | 0.01 | 5 | 0.7 | 0.9 | N/A | N/A | N/A |
| | | SGDM | 0.1 | 0 | 0 | 0.9 | N/A | N/A | N/A |
| | | $\text{SGD}_C$ | 0.1 | 5 | 0.7 | N/A | N/A | N/A | N/A |
| | | SGD | 0.1 | 0 | 0 | N/A | N/A | N/A | N/A |
| | ConvEncoder | $\text{Adam}_C$ | 0.0001 | 5 | 0.7 | N/A | 0.99 | 0.999 | N/A |
| | | Adam | 0.001 | 0 | 0 | N/A | 0.99 | 0.999 | N/A |
| | | $\text{RMSprop}_C$ | 0.0001 | 5 | 0.7 | N/A | N/A | N/A | 0.99 |
| | | RMSprop | 0.001 | 0 | 0 | N/A | N/A | N/A | 0.99 |
| | | $\text{SGDM}_C$ | 0.01 | 5 | 0.7 | 0.9 | N/A | N/A | N/A |
| | | SGDM | 0.1 | 0 | 0 | 0.9 | N/A | N/A | N/A |
| | | $\text{SGD}_C$ | 0.1 | 5 | 0.7 | N/A | N/A | N/A | N/A |
| | | SGD | 0.1 | 0 | 0 | N/A | N/A | N/A | N/A |
| | BLSTMEncoder | $\text{Adam}_C$ | 0.0001 | 5 | 0.7 | N/A | 0.99 | 0.999 | N/A |
| | | Adam | 0.001 | 0 | 0 | N/A | 0.99 | 0.999 | N/A |
| | | $\text{RMSprop}_C$ | 0.0001 | 5 | 0.7 | N/A | N/A | N/A | 0.99 |
| | | RMSprop | 0.001 | 0 | 0 | N/A | N/A | N/A | 0.99 |
| | | $\text{SGDM}_C$ | 0.01 | 5 | 0.7 | 0.9 | N/A | N/A | N/A |
| | | SGDM | 0.1 | 0 | 0 | 0.9 | N/A | N/A | N/A |
| | | $\text{SGD}_C$ | 0.1 | 5 | 0.7 | N/A | N/A | N/A | N/A |
| | | SGD | 0.1 | 0 | 0 | N/A | N/A | N/A | N/A |

using both aggregation methods, we found that $\text{SGD}_C$ and $\text{SGDM}_C$ demonstrated better validation performance using `sum` whereas $\text{RMSprop}_C$ and $\text{Adam}_C$ performed better using `mean`. This is likely due to both RMSprop and Adam having adaptive learning rates, allowing them to be more robust to changes in the scaling of the gradients.

## F.2 CONVERGENCE IN CONVEX TASKS

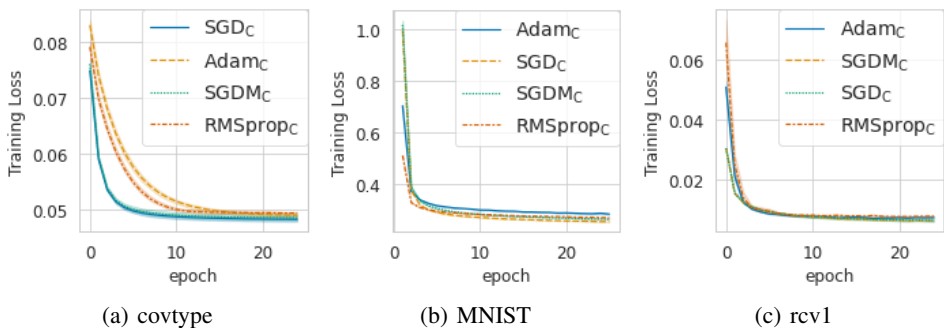

(a) covtype  (b) MNIST  (c) rcv1

Figure 7: The training on the three different convex datasets show that the $_C$ variants of the base optimizers empirically converge.

To empirically validate our proof of convergence for the proposed methods on convex loss surfaces, we train all $_C$ optimizer variants on three tasks with Logistic Regression: $\ell_2$-regularized binary classification with *rcv1* (Lewis et al., 2004), $\ell_2$-regularized multi-class classification on *covtype* (Dua and Graff, 2017), and non-regularized multi-class classification on MNIST. Figure 7 shows training losses for all optimizers converging towards a minimal training loss.

## F.3 BUFFER STALENESS BOUND

We present additional results demonstrating the boundedness of the staleness of stored gradients (Figure 8).

## F.4 ADDITIONAL ABLATION EXPERIMENTS

We present additional side-by-side comparisons of optimizers in the vein of §6 (Figure 9).

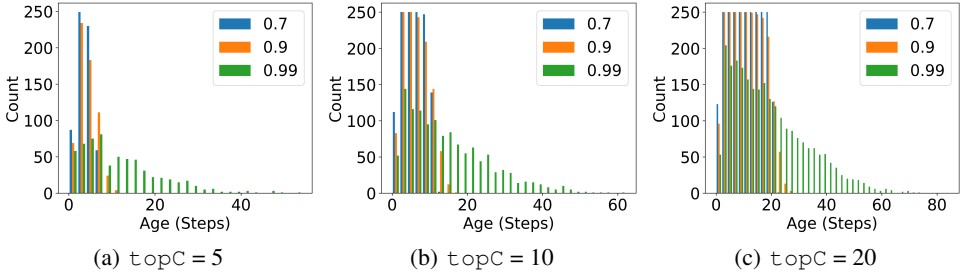

(a) `topC = 5`  (b) `topC = 10`  (c) `topC = 20`

Figure 8: Histograms showing ages of the buffer as recorded at the end of every epoch, across five seeds, for Logistic Regression on MNIST.

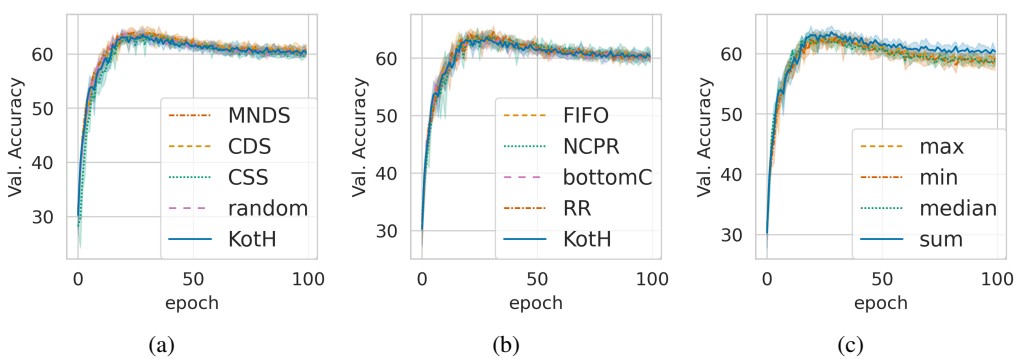

(a)  (b)  (c)

Figure 9: Additional ablation experiments performed on CIFAR-10

## F.5 ROBUSTNESS TO NOISE

As $_C$ variants retain larger gradients during training, and noisy data points may have gradients with higher values of $\ell_2$-norm, we experiment with the $_C$ variants by training Logistic Regression on a synthetic binary classification dataset where the labels are perturbed with probability $\mathcal{P}_{noise}$.

We sample 500 data points from a 7-dimensional hypercube with the class means separated by 1 unit. The data is split by 80/20 for training and evaluation. We train the model with the optimizers for 20 epochs on a fixed learning rate of $1 \times 10^{-3}$, `topC = 5`, `decay = 0.5` and other parameters from the base model set to their defaults.

We compare the performance of base method and its $_C$ variant to study anomalies when the $_C$ variants are exposed to noise (Figure 10). We observe that the models do not behave any differently than the base methods showcasing that the noise in the dataset does not affect the workings of the $_C$ variants even with a small value for `topC`. This could be attributed to the `decay` parameter of the optimizer that linearly scales down the priority of the older gradients in the buffer leading to them getting replaced.

To construct the synthetic data set for the experiments, we used sklearn's `make_classification` method. The specific hyperparameters used to construct the dataset:

- n_samples = 500, n_features = 10,
- n_informative=7, n_redundant=0, n_repeated=0,
- n_classes=2, n_clusters_per_class=1,
- weights=class_imbalance, flip_y=noise, class_sep= 0.5,
- hypercube=True, shift=0.4, scale=1.0, shuffle=True, random_state=1403

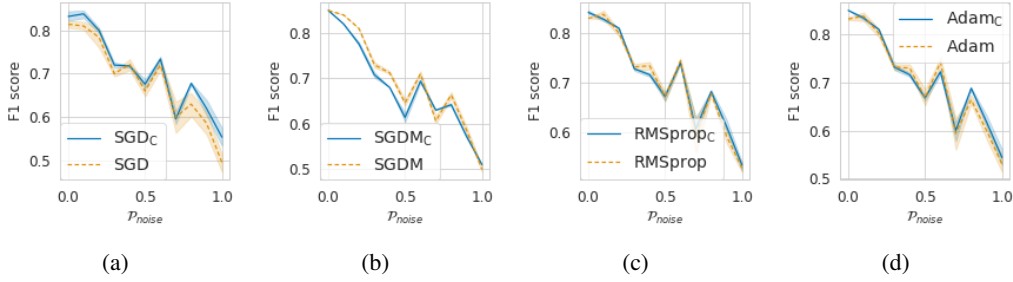

|     |     |     |     |
| :-: | :-: | :-: | :-: |
| (a) | (b) | (c) | (d) |

Figure 10: $\mathcal{P}_{noise}$ is increased from 0 to 1 in increments of 0.1 indicating 10% more injection of noise. The proposed variants did not show any anomaly on a noise induced dataset. For the experiments on the decay the `topC` was 5 and `decay` 0.5. The plots are averaged over 5 different runs with different seeds.

## F.6    DECAY

The optimizers' sensitivity to `decay` (Figure 11) had similar trend in other experiments, where the optimizers showed better performance when `decay` was away from 0.99.

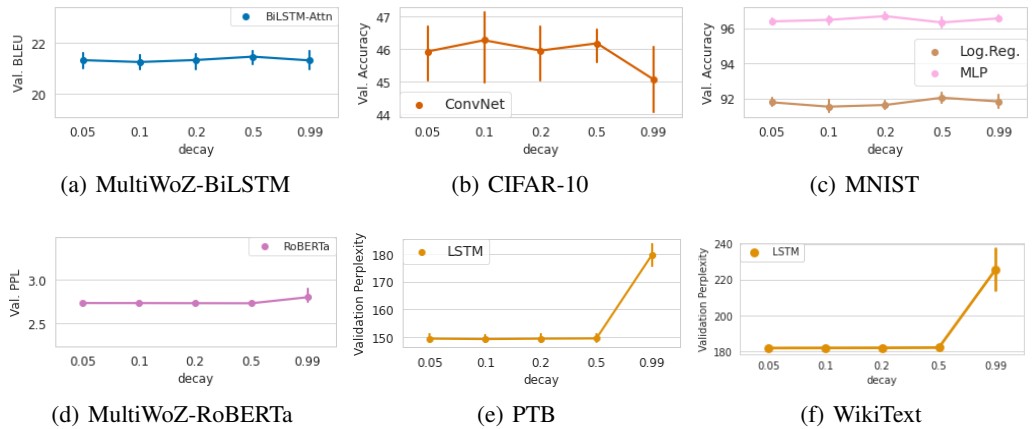

Figure 11: Experiments on varying `decay` that across tasks. Graphs in the top row measure BLEU score or accuracy (where a higher value is desired) and those in the bottom row measure perplexity (where a lower value is desired)

## F.7    TOPC

The optimizers' sensitivity to `topC` (Figure 12) had a similar trend to other experiments, where the optimizers showed better performance for lower values of `topC`.

## F.8    $g_t$ VS $g_c$

As a follow-up experiment to validate the non-informative gradients with higher values of `topC`, we observe the trend in the difference between `average`($\|g_c\|_2$) and $\|g_t\|_2$ for different values of `topC`. We see that the gradients at each step get lower as `topC` increases. This could be because of the stochasticity in `aggr` when computed with fewer gradients, which incidentally allow models to converge better. Although storing all of the past gradients in memory has theoretical advantages, in practice we observe that lower `topC` provides better training signal through $g_c$ than $g_t$ (Figure 13(a),13(b)). This explains the slightly lower performance of `topC` =100 across models in SNLI task and the nonexistent to marginal improvements in MultiWoZ dataset with RoBERTa-Base model.

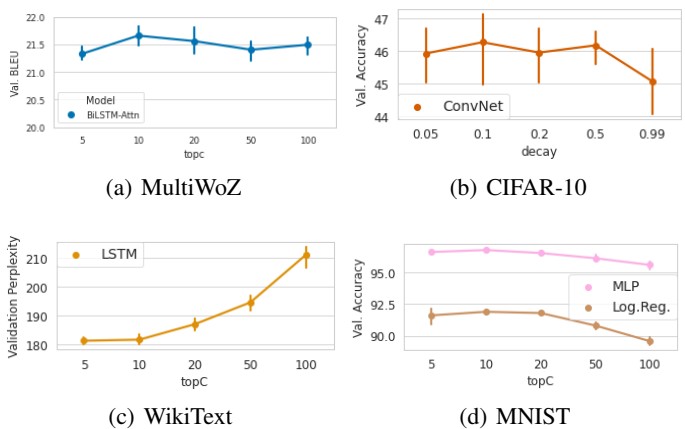

(a) MultiWoZ

(b) CIFAR-10

(c) WikiText

(d) MNIST

Figure 12: Experiments on varying `topC` that across tasks. Top row: (a) MultiWoz - BiLSTM (BLEU Score) (b) CIFAR-10 - Convnet (Accuracy). Bottom row: (c) WikiText - LSTM (Perplexity) (d) MNIST - Log. Reg. and MLP (Accuracy).

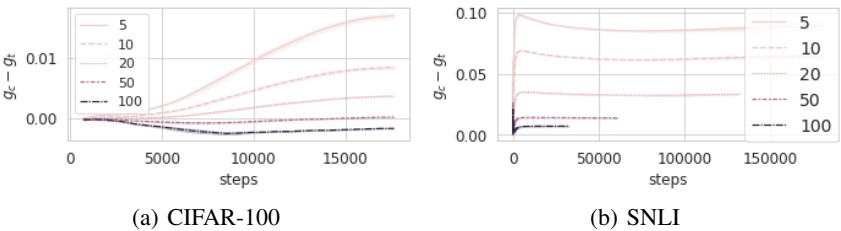

(a) CIFAR-100

(b) SNLI

Figure 13: The difference between $\texttt{average}(||g_c||_2)$ and $||g_t||_2$ shown in plots (a) and (b) indicate that increasing `topC` could hurt the performance. These plots correlate to experiments in §6

The plots of $g_t$ vs $g_c$ is crucial to the results in that they provide an explanations for the faster convergence (as well as smaller improvements) of the $_C$ variants over the base methods (Figure 14). Since the parameter updates are reminded of large displacement of the gradients via the `aggr` method, the model updates the parameters more frequently. In the cases where $g_c$ is higher than $g_t$, we observe better performance and no improvements in cases where the difference is not as significant.

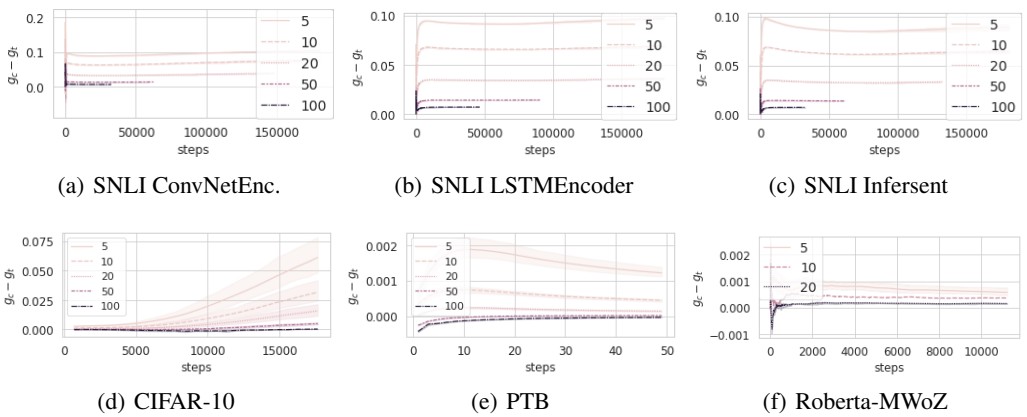

(a) SNLI ConvNetEnc.  (b) SNLI LSTMEncoder  (c) SNLI Infersent

(d) CIFAR-10  (e) PTB  (f) Roberta-MWoZ

Figure 14: Additional results on $g_t$ vs $g_c$, explaining rationale behind the better performances of lower values of `topC` across all the experiments. As the variance in `aggr` get diminished with increase in `topC` the model does not gain much with increase in `topC` .

Further, we observe the significance of the gradients as defined in $\ell_2$-norm diminishes as `topC` increases. This indicates that the optimizer needs only to store only a small set of critical gradients in order to improve the performance without incurring much time and memory overhead. Employing lazy updates to cut down on the time is a useful future direction of research.

## F.9 DETAILS OF ABLATION EXPERIMENTS

We detail the various ablation experiments whose results are presented in §6.

- King of the Hill sampling: The default method for Critical Gradients. Incoming gradients are added if their norm is larger than the priority of the smallest gradient, which is subsequently removed.

- Smallest gradients (or "`bottomC`"): Replaces the max-heap buffer with a min-heap

- First-In-First-Out (FIFO): Replaces the max-heap (priority queue) buffer with a queue which begins to dequeue when it reaches a capacity of `topC` .

- "Coin-Toss"/"random": an incoming gradient is added to the buffer or not with equal probability.

- Mean-Norm Diversity Sampling (MNDS): Incoming gradients are probabilistically added to the buffer with likelyhood proportional to its difference from the mean of the norms of gradients in the buffer.

- Cosine Diversity Sampling (CDS): Incoming gradients are probabilistically added to the buffer with likelyhood *antiproportional* to its cosine similarity (normalized dot-product of flattened vectors).

- Cosine Similarity Sampling (CSS): Incoming gradients are probabilistically added to the buffer with likelyhood *proportional* to its cosine similarity.

- Random Replacement (RR): When the buffer is full and a new gradient is added, the gradient which gets removed gets selected at random.

- Norm-Controlled Probabilistic Replacement (NCPR): When the buffer is full and a new gradient is added, the gradient $g_k$ which gets removed gets selected with probability $\frac{||g_k||_2}{\sum_{g_i \in \mathbf{g_c}} ||g_i||_2}$.

## G    TEST PERFORMANCE

The complete test performance of the models is reported in Table 10.

## H    REPRODUCIBILITY CHECKLIST

As per the prescribed Reproducibility Checklist, we provide the information of the following:

- *A clear description of the mathematical setting, algorithm and/or model*: We provide details of models used in §D
- *Submission of source code*: Source code for the proposed optimizers and its variants is provided as a zip. The code used to train the models on the different data sets are open source GitHub repositories. Other codes developed for the project are included in the zip.
- *Description of the computing infrastructure used*: We used 50 NVIDIA V100 32GB GPUs in parallel hyper parameter search over the grid using wandb and submitit packages. For the final runs we used 1 NVIDIA V100 32 GB GPUs for every seed of every model.
- *Average runtime for each approach*: The approximate training time for our use of $\mathrm{Adam_C}$ accross tasks is reported in §D.2.
- *Explanation of evaluation metrics used, with links to code*: The metrics used for evaluation of the models are the popular ones. For the ease of readers citations for the metrics are included in the paper.
- *Relevant statistics of the datasets used*: We provide the statistics of the datasets used in C.
- *Explanation of any data that were excluded, and all pre-processing steps*: We train on a fraction of the covtype and rcv1 datasets instead of using the entire data. We sampled 5000 datapoints at random with seed set to 100.
- *Link to downloadable version of data*: The data sets used in the paper are from public repositories. Links to the paper that proposes the data sets is included in the README.md files in the submitted repository.

Table 10: Complete test results on the different tasks.

| Model | Dataset | Metric | Optimizer | Performance |
|---|---|---|---|---|
| RoBERTa | MWoZ | PPL | Adam | $2.57 \pm 0.0$ |
| RoBERTa | MWoZ | PPL | $\text{Adam}_C$ | $2.41 \pm 0.0$ |
| RoBERTa | MWoZ | PPL | RMSprop | $2.56 \pm 0.01$ |
| RoBERTa | MWoZ | PPL | $\text{RMSprop}_C$ | $2.42 \pm 0.0$ |
| RoBERTa | MWoZ | PPL | SGDM | $2.46 \pm 0.0$ |
| RoBERTa | MWoZ | PPL | $\text{SGDM}_C$ | $2.4 \pm 0.01$ |
| RoBERTa | MWoZ | PPL | SGD | $2.62 \pm 0.01$ |
| RoBERTa | MWoZ | PPL | $\text{SGD}_C$ | $2.41 \pm 0.0$ |

| Model | Dataset | Metric | Optimizer | Performance |
|---|---|---|---|---|
| BiLSTM | MWoZ | BLEU | Adam | $20.64 \pm 0.19$ |
| BiLSTM | MWoZ | BLEU | $\text{Adam}_C$ | $20.94 \pm 0.26$ |
| BiLSTM | MWoZ | BLEU | RMSprop | $21.3 \pm 0.41$ |
| BiLSTM | MWoZ | BLEU | $\text{RMSprop}_C$ | $21.21 \pm 0.22$ |
| BiLSTM | MWoZ | BLEU | SGDM | $19.31 \pm 0.27$ |
| BiLSTM | MWoZ | BLEU | $\text{SGDM}_C$ | $19.58 \pm 0.47$ |
| BiLSTM | MWoZ | BLEU | SGD | $14.26 \pm 0.61$ |
| BiLSTM | MWoZ | BLEU | $\text{SGD}_C$ | $16.35 \pm 0.24$ |

| Model | Dataset | Metric | Optimizer | Performance |
|---|---|---|---|---|
| LSTM | PTB | PPL | Adam | $153.54 \pm 1.65$ |
| LSTM | PTB | PPL | $\text{Adam}_C$ | $133.28 \pm 0.61$ |
| LSTM | PTB | PPL | RMSprop | $141.83 \pm 0.57$ |
| LSTM | PTB | PPL | $\text{RMSprop}_C$ | $130.63 \pm 0.64$ |
| LSTM | PTB | PPL | SGDM | $139.93 \pm 0.64$ |
| LSTM | PTB | PPL | $\text{SGDM}_C$ | $132.22 \pm 1.1$ |
| LSTM | PTB | PPL | SGD | $386.57 \pm 18.61$ |
| LSTM | PTB | PPL | $\text{SGD}_C$ | $295.33 \pm 2.96$ |

| Model | Dataset | Metric | Optimizer | Performance |
|---|---|---|---|---|
| LSTM | WikiText | PPL | Adam | $170.3 \pm 1.23$ |
| LSTM | WikiText | PPL | $\text{Adam}_C$ | $169.23 \pm 1.06$ |
| LSTM | WikiText | PPL | RMSprop | $180.42 \pm 1.64$ |
| LSTM | WikiText | PPL | $\text{RMSprop}_C$ | $172.15 \pm 1.63$ |
| LSTM | WikiText | PPL | SGDM | $166.82 \pm 2.24$ |
| LSTM | WikiText | PPL | $\text{SGDM}_C$ | $156.75 \pm 1.32$ |
| LSTM | WikiText | PPL | SGD | $461.84 \pm 12.68$ |
| LSTM | WikiText | PPL | $\text{SGD}_C$ | $356.88 \pm 8.34$ |

| Model | Dataset | Metric | Optimizer | Performance |
|---|---|---|---|---|
| InferSent | SNLI | Accuracy | Adam | $78.42 \pm 0.26$ |
| InferSent | SNLI | Accuracy | $\text{Adam}_C$ | $79.03 \pm 0.18$ |
| InferSent | SNLI | Accuracy | RMSprop | $77.91 \pm 0.35$ |
| InferSent | SNLI | Accuracy | $\text{RMSprop}_C$ | $79.07 \pm 0.28$ |
| InferSent | SNLI | Accuracy | SGDM | $79.15 \pm 0.3$ |
| InferSent | SNLI | Accuracy | $\text{SGDM}_C$ | $78.52 \pm 0.26$ |
| InferSent | SNLI | Accuracy | SGD | $78.3 \pm 0.16$ |
| InferSent | SNLI | Accuracy | $\text{SGD}_C$ | $78.9 \pm 0.57$ |

| Model | Dataset | Metric | Optimizer | Performance |
|---|---|---|---|---|
| ConvNet | SNLI | Accuracy | Adam | $75.52 \pm 0.0$ |
| ConvNet | SNLI | Accuracy | $\text{Adam}_C$ | $77.77 \pm 0.28$ |
| ConvNet | SNLI | Accuracy | RMSprop | $74.69 \pm 0.36$ |
| ConvNet | SNLI | Accuracy | $\text{RMSprop}_C$ | $77.49 \pm 0.24$ |
| ConvNet | SNLI | Accuracy | SGDM | $76.79 \pm 0.68$ |
| ConvNet | SNLI | Accuracy | $\text{SGDM}_C$ | $78.06 \pm 0.45$ |
| ConvNet | SNLI | Accuracy | SGD | $78.5 \pm 0.1$ |
| ConvNet | SNLI | Accuracy | $\text{SGD}_C$ | $78.59 \pm 0.24$ |

| Model | Dataset | Metric | Optimizer | Performance |
|---|---|---|---|---|
| LSTMEnc | SNLI | Accuracy | Adam | $76.79 \pm 0.38$ |
| LSTMEnc | SNLI | Accuracy | $\text{Adam}_C$ | $77.3 \pm 0.09$ |
| LSTMEnc | SNLI | Accuracy | RMSprop | $76.37 \pm 0.36$ |
| LSTMEnc | SNLI | Accuracy | $\text{RMSprop}_C$ | $77.48 \pm 0.11$ |
| LSTMEnc | SNLI | Accuracy | SGDM | $77.6 \pm 0.22$ |
| LSTMEnc | SNLI | Accuracy | $\text{SGDM}_C$ | $76.69 \pm 1.04$ |
| LSTMEnc | SNLI | Accuracy | SGD | $74.92 \pm 1.32$ |
| LSTMEnc | SNLI | Accuracy | $\text{SGD}_C$ | $76.81 \pm 0.59$ |

| Model | Dataset | Metric | Optimizer | Performance |
|---|---|---|---|---|
| ConvNet | CIFAR100 | Accuracy | Adam | $54.71 \pm 0.55$ |
| ConvNet | CIFAR100 | Accuracy | $\text{Adam}_C$ | $53.55 \pm 0.42$ |
| ConvNet | CIFAR100 | Accuracy | RMSprop | $54.35 \pm 0.5$ |
| ConvNet | CIFAR100 | Accuracy | $\text{RMSprop}_C$ | $52.93 \pm 0.57$ |
| ConvNet | CIFAR100 | Accuracy | SGDM | $56.03 \pm 0.49$ |
| ConvNet | CIFAR100 | Accuracy | $\text{SGDM}_C$ | $59.07 \pm 0.2$ |
| ConvNet | CIFAR100 | Accuracy | SGD | $55.37 \pm 0.39$ |
| ConvNet | CIFAR100 | Accuracy | $\text{SGD}_C$ | $58.0 \pm 0.37$ |

| Model | Dataset | Metric | Optimizer | Performance |
|---|---|---|---|---|
| ConvNet | CIFAR10 | Accuracy | Adam | $64.26 \pm 0.47$ |
| ConvNet | CIFAR10 | Accuracy | $\text{Adam}_C$ | $63.72 \pm 0.37$ |
| ConvNet | CIFAR10 | Accuracy | RMSprop | $63.9 \pm 0.33$ |
| ConvNet | CIFAR10 | Accuracy | $\text{RMSprop}_C$ | $64.02 \pm 0.78$ |
| ConvNet | CIFAR10 | Accuracy | SGDM | $64.7 \pm 0.69$ |
| ConvNet | CIFAR10 | Accuracy | $\text{SGDM}_C$ | $63.68 \pm 0.77$ |
| ConvNet | CIFAR10 | Accuracy | SGD | $63.9 \pm 0.63$ |
| ConvNet | CIFAR10 | Accuracy | $\text{SGD}_C$ | $64.89 \pm 0.66$ |

