# OpenReview forum: "Memory Augmented Optimizers for Deep Learning"
_ICLR.cc/2022/Conference — ICLR 2022 Poster_

### Official Review · Reviewer_v5GJ · 2021-11-01

**Correctness:** 4
**Technical Novelty And Significance:** 3
**Empirical Novelty And Significance:** 3
**Recommendation:** 8
**Confidence:** 4

**Main Review:**

Strengths:

- analysis on strongly convex smooth objectives
- good empirical evaluation
- strong results on evaluated datasets

Weaknesses

- nowhere near SotA validation on CIFAR 10/100? Validation should be around 75% looking at wideresnets, results, while this is an optimisation paper, I feel like this makes it hard to compare/more vulnerable to accidental cherry picking
- architectures only visible in supplimentary, and appendix listing them for analysis would help the reader to have all information in one place
- strongly convex smooth is a strong assumption to place on convergence, although the empirical performance alleviates this to some degree
- I'm missing some comparison to other discussions of gradient staleness e.g. from the Asychronous Distributed SGD community http://proceedings.mlr.press/v80/damaskinos18a/damaskinos18a.pdf

Questions:

- do you have any explanations of the cases where you regress in performance/speed? violation of the assumptions, high gradient variance or anything else?

**Summary Of The Paper:**

The paper proposes an optimisation framework that uses K previous gradients stored in a buffer to compute an aggregated proxy-gradient for the  use in other gradient frameworks (studied are SGD, SGDM, RMSProp, Adam) and evaluates it on CIFAR 10/100 as well as NLP tasks. The paper also presents a convergence proof on strongly convex smooth objectives. In almost all empirical evaluations the method shows an improvement in terms of iteration and performance and ablations are performed to show strong robustness to hyperparameters introduced.

**Summary Of The Review:**

Overall, I like this paper as a simple to implement empirical improvement to other algorithm that just "makes sense": very large gradients should probably have an impact for a longer time BUT not all at once so as to not dominate fully.  It gives good but not overwhelming empirical evidence (difficult to compare to SotA due to architecture choice and different tuning) and some theoretical justification (albeit not applicable to the settings evaluated).

---

> ### Author Response · Authors · 2021-11-22
> **Official Response to Reviewer v5GJ**
>
> We thank the reviewer for their suggestions and comments. We address the questions raised in the review in detail below:
>
> **Review**: nowhere near SotA validation on CIFAR 10/100? Validation should be around 75% looking at wideresnets, results, while this is an optimisation paper, I feel like this makes it hard to compare/more vulnerable to accidental cherry picking.\
> **Response:** We used a CNN with 5 and 9 convolutional layers respectively for CIFAR-10 and CIFAR-100. These are standard architectures, used in some of the popular research works like [1,2]. We would like to highlight that CIFAR-100 performances reported is in similar range as in  [2].
>
> [1] Paul, M., Ganguli, S. and Dziugaite, G.K., 2021. Deep Learning on a Data Diet: Finding Important Examples Early in Training. arXiv preprint arXiv:2107.07075.
> [2] Schmidt, R.M., Schneider, F. and Hennig, P., 2021, July. Descending through a crowded valley-benchmarking deep learning optimizers. In International Conference on Machine Learning (pp. 9367-9376). PMLR.
> ___
>
> **Review:** strongly convex smooth is a strong assumption to place on convergence, although the empirical performance alleviates this to some degree.\
> **Response:** Like the reviewer pointed out, our extensive empirical analysis on several non-convex problems shows that the critical gradients indeed converge faster in non-convex settings. Proving convergence theoretically in non-convex setting is a very interesting future work.
> ___
>
> **Review:** I'm missing some comparison to other discussions of gradient staleness e.g. from the Asychronous Distributed SGD community
> http://proceedings.mlr.press/v80/damaskinos18a/damaskinos18a.pdf \
> **Response:**  We appreciate the reference. We will include it in our final version along with a discussion. Whereas gradient staleness is typically leveraged in distributed stochastic gradient methods to introduce fault tolerance and or reduce communication/synchronization overhead, our motivation is in serial optimization. It would be very interesting to explore version of critical gradients in distributed asynchronous settings to significantly reduce the communication overhead of large-scale distributed optimization. We plan to explore this in future work.
> ___
>
> **Review:** do you have any explanations of the cases where you regress in performance/speed? violation of the assumptions, high gradient variance or anything else?\
> **Response:** We do not have a clear reason. But, a hypothesis we formulated have from experimenting with testing the optimizers on different loss surfaces is that the C-optimizers (our method) gains acceleration over the base optimizers. The acceleration obtained in _some_ scenarios overshoots the descent leading to the updates missing the local optima in some pathological those cases. Though the hyperparameters – decay and topC – allows control to a greater degree,  more analysis is needed to understand the failure scenarios.
> ___
> Please let us know if you have further clarifications. We would be happy to address them during the discussion period.

---

### Official Review · Reviewer_tj5C · 2021-11-03

**Correctness:** 3
**Technical Novelty And Significance:** 3
**Empirical Novelty And Significance:** 4
**Recommendation:** 6
**Confidence:** 4

**Main Review:**

Strengths:
- A general framework which can accommodate existing first order optimizers to trade off memory with faster convergence.
- Convergence proof for smooth strongly convex functions
- Solid set of experiments
  - Sensitivity to hparams
  - Alternatives to l_2 norm for deeming gradients critical


Areas of improvements and questions:
- Either make figure 1 vertical or at least make the text vertical so that its easier to read.
- In Figure 2, I see that several tasks degrade convergence. Is there a way to make sure this method does no worse than before? I will increase my score if this is done.

**Summary Of The Paper:**

Bridging the advances from theoretical and empirical side, this paper proposes a framework for memory augmented optimizers, a strategy which can improve any off-the-shelf first order optimizers trading off increased memory with faster convergence.

**Summary Of The Review:**

Overall good paper, some areas of improvements

---

> ### Author Response · Authors · 2021-11-17
> **Official Response to Reviewer tj5C**
>
> We appreciate the discussion posted by the reviewer and address them in detail below:\
> **Review:** In Figure 2, I see that several tasks degrade convergence. Is there a way to make sure this method does no worse than before? I will increase my score if this is done.
>
> **Response:** The critical gradient methods are a generalization of the base optimizers, therefore it is always possible to choose hyper-parameters to recover the base performance. Figure 2 compares every base optimizer with its critical gradient counterpart for each task. However, a better comparison would be to compare the best performing base optimizer with the best performing critical gradient optimizer for the given task. We summarize the same from Table 10, here, below:\
> | **Task**     | **Model**     | **Best-C (Our method)** | **Best-Base** |         **Comment**          |\
> | -------- | --------- | ------ | --------- | ----------------- |\
> | MWoZ     | RoBERTa   | **2.4**    | 2.46      | Lower the better  |\
> | MWoZ     | BiLSTM    | 21.21  | **21.3**      | Higher the better |\
> | PTB      | LSTM      | **132.22** | 141.83    | Lower the better  |\
> | WikiText | LSTM      | **156.76** | 166.82    | Lower the better  |\
> | SNLI     | InferSent | **79.15**  | 79.07     | Higher the better |\
> | SNLI     | ConvNet   | **78.59**  | 78.5      | Higher the better |\
> | SNLI     | LSTMEnc   | 77.48  | **77.6**     | Higher the better |\
> | CIFAR10  | ConvNet   | **64.89**  | 64.26     | Higher the better |\
> | CIFAR100 | ConvNet   | **59.07**  | 56.03     | Higher the better |
>
>  In fact,  when the results are aggregated, we see that the base optimizers only perform better (marginally) in two of the nine scenarios. In these two scenarios, choosing buffersize to be 1 and decay to be 0 will recover base performance.
> ___
>
> Please let us know if you have further clarifications. We would be happy to address them during the discussion period.

---

### Official Review · Reviewer_XUqb · 2021-11-03

**Correctness:** 4
**Technical Novelty And Significance:** 3
**Empirical Novelty And Significance:** 3
**Recommendation:** 6
**Confidence:** 3

**Main Review:**

The critical gradient strategy seems fairly interesting and a promising way to accelerate training. The paper was fun to read and well written. The experimental evaluation is fairly comprehensive and has interesting analysis. The main concerns I have are:
* Although the convergence speed in terms of iterations is faster, the wall clock time relative to baselines especially with large buffer sizes seems much higher Appendix D2 Table 4. The overhead seems to stem from the book keeping, replacement and aggregation strategy. Is there a convergence speed comparison Figure 2 with wall clock times as opposed to number of gradient updates?
* In Analysis there is nice study on the staleness of the gradients in the buffer. Can the staleness bound just be enforced instead of relying on it being empirically the case?
* If the critical gradients are finally aggregated either via sum or averaging there can be easier strategies for doing the same aggregation in a more online fashion without additional memory. Especially given that there is a decay term when doing gradient selection for replacement. Was there an attempt to try more online approaches?


**Summary Of The Paper:**

The paper proposes a framework for memory-augmented gradient descent optimizers which only keep a limited a limited buffer of gradient history and can be integrated with existing optimizers. The method proposed in the paper is centered around "critical gradients" with large l2 norm and only retaining them in the limited buffer. Experimental results show the faster convergence in terms of iterations compared to standard SGD algorithms without critical gradient buffers.


**Summary Of The Review:**

The proposed method of only storing critical gradients in a limited size memory buffer seems interesting. The experimental evaluation is comprehensive. However, the method only seems to improve in terms of gradient steps but not wall clock time to converge due to bookkeeping overheads.

---

> ### Author Response · Authors · 2021-11-18
> **Official Response to Reviewer XUqb**
>
> We thank the reviewer for their comments and suggestions. We address them in detail below:\
> **Review:** Is there a convergence speed comparison Figure 2 with wall clock times as opposed to number of gradient updates?\
> **Response:** Table 4 provides the per-epoch clock time for the optimizers. We would like to highlight that in all of the tasks we found topC=5/10 to be the best performing hyperparam for our method. For topC=5/10, even though our method is currently lagging behind baseline optimizers in terms of clock-time per epoch, we train in much fewer epochs (Figure 2).
> Further, we would like to highlight that our current implementation of the critical gradient optimizer is not optimal with respect to switching between CPU/GPU operations and the data structure used for the buffer. As an update to the existing implementation, we are working to improve the implementation to gain speedup even when the value of topC is higher.
> ___
> **Review:** In Analysis there is nice study on the staleness of the gradients in the buffer. Can the staleness bound just be enforced instead of relying on it being empirically the case? \
> **Response:** Figure 5 (b) compares the different gradient dropout strategies; FIFO strategy is similar to what you suggest. In this case we see the different strategies having similar performances in language modeling task. We intend to conduct an elaborate empirical study in a future work to understand this phenomenon.
> ___
> **Review:** Was there an attempt to try more online approaches?\
> **Response:** The paper conveys that having an explicit memory (as in Adam_C) instead of completely online gradient aggregation (as in Adam) helps in accelerating the convergence of the optimization process. While reducing the memory is non-trivial, we can still reduce the compute and bring memory augmented optimizers close to compute time of online optimizers by smart implementations.
> ___
> Please let us know if you have further clarifications. We would be happy to address them during the discussion period.

---

### Official Review · Reviewer_KE2X · 2021-11-03

**Correctness:** 3
**Technical Novelty And Significance:** 2
**Empirical Novelty And Significance:** 3
**Recommendation:** 5
**Confidence:** 4

**Main Review:**

Pros
* The selection of the critical gradients directly depends on the $l_2$ norm of the gradients and the aggregation function is simple enough, so the idea itself looks reasonable and the implementation of the algorithm is straightforward. Furthermore, the optimizer could be combined with other popular first-order optimizers which can make it more useful.
* The related work section is well-written and the number of citations is adequate. I feel comfortable reading through the paper and the points are obvious to me. The comparison between the proposed method and other algorithms on memory requirements and performance improvement is clear.
* The experiment results are strong. Figure 2 gives a nice summary about how the proposed memory-augmented optimizer influences performance of the original optimizer.

Cons
* Although it seems natural to choose gradients of large norm as the critical ones. I did not find any motivation or explanation for that. There are several related works trying to utilize the information of historical gradients, but it is not clear how saving the critical gradients by looking at their norm helps improve the performance. More specifically, In Figure 5 (a) and (b), it seems the choices of selection techniques and replacement methods do not influence the results a lot. I would be rather interested if the author could give some insights about the reason why such selection mechanisms work well.

* The theoretical proof looks merely fine to me. It is not clear how the choice of $K$ will influence the bound. It would be more obvious and convincing if the authors could give some examples rather than simply claim it may be possible to accelerate convergence rates when $K > 1$. I guess such acceleration might depend on specific conditions and these conditions could show some improvements and limitations of the proposed algorithm.

Comments
* Maybe there is a limitation for the number of pages, but I personally feel it will make the paper look more complete if Algorithm 1 goes to the main body.
* The description of $A_t$ at the bottom of page 13 seems to be wrong. Shouldn't the blocks of identity matrices be moved one block to their left?

**Summary Of The Paper:**

The paper proposes a memory-augmented optimizer to make use of the history information of the gradients. The motivation is that the aggregation of past gradients nudges parameter updates in a better direction. However, it remains to answer how much historical information of the gradients is sufficient. The authors study and analyze the memory requirement of several common optimization algorithms. Based on that, the authors propose a new optimizer that retains a limited gradient history and those gradients are selected by their importance. This method could be built upon existing popular optimizers like Adam and SGD and enjoys accelerated convergence and improved performance.

**Summary Of The Review:**

In general, the paper is well-written and the experimental results look strong. However, the lack of clear motivation and explanation of the proposed method downgrades my rating to the paper. Furthermore, the theoretical part could be more detailed as the current one does not provide a convincing improvement over vanilla SGD method. I hope the authors will answer these questions in the future.

---

> ### Author Response · Authors · 2021-11-17
> **Official Response to Reviewer KE2X**
>
> We appreciate the comments and clarifications raised by the reviewer and would like to address them in detail.
>
> **Review:** Although it seems natural to choose gradients of large norm as the critical ones. I did not find any motivation or explanation for that. I would be rather interested if the author could give some insights about the reason why such selection mechanisms work well.
>
> **Response:** One intuitive motivation is through data-pruning. Recent work [1] has identified that samples with larger gradient norms can be used to identify a smaller set of training data that is important for generalization, and thereby prune the data as a pre-processing step. The memory buffer in critical gradient variants can be seen as a soft version of data-pruning in that the buffer retains gradients through the heuristic, thereby performing updates using only the important gradients throughout the entire training, without discarding data in a pre-processing step. Also, we will include this explanation in our revised version.
>
> [1] Paul, M., Ganguli, S. and Dziugaite, G.K., 2021. Deep Learning on a Data Diet: Finding Important Examples Early in Training. arXiv preprint arXiv:2107.07075.
> ___
>
> **Review:** In Figure 5 (a) and (b), it seems the choices of selection techniques and replacement methods do not influence the results a lot.
>
> **Response:** The motivation for this ablation experiment was to demonstrate the potential of memory augmentation and the importance of using an ensemble of gradients. We would like to highlight that the ablation in Figure 5(a) and (b) was performed on a single task and cannot necessarily be generalized.
> ___
>
> **Review:** The theoretical proof looks merely fine to me. It is not clear how the choice of K will influence the bound. It would be more obvious and convincing if the authors could give some examples rather than simply claim it may be possible to accelerate convergence rates when K>1. I guess such acceleration might depend on specific conditions and these conditions could show some improvements and limitations of the proposed algorithm.
>
> **Response:** Yes indeed, our theoretical proof provides a convergence guarantee for a wide variety of critical gradient methods and aggregation strategies, hence theoretical acceleration will depend on the specific conditions. We conduct a broad empirical analysis of different critical gradient variants for a wide range of non-convex deep learning problems spanning several modalities (text, vision, etc.) to show faster convergence and improved generalization in almost all of the tasks.
> ___
>
> Please let us know if you have further clarifications. We would be happy to address them during the discussion period.

---

### Decision · Program_Chairs · 2022-01-20

**Decision:**

Accept (Poster)

**Comment:**

The paper proposes a general method to enhance the performance of first-order optimizers. The main idea is to use a memory buffer to maintain a limited set of critical gradients from recent history. Namely, gradients with large l2 norm. The paper includes a convergence proof on strongly convex smooth objectives. Experimental results are reported for several architectures in vision and language tasks. When integrated with several commonly used optimizers (SGD, SGDM, RMSProp, Adam), the method shows an improvement in terms of learning speed as well as improved performance (in almost all cases). Several ablations are performed to show strong robustness to hyperparameters introduced.

The paper is very well written and easy to follow. The proposed method is simple and effective. The empirical evaluation is strong and the ablation studies exhaustive and convincing, as pointed out by all four reviewers.

The authors provided a solid rebuttal addressing many questions raised by the reviewers.

Reviewer KE2X is the only reviewer recommending to reject the paper, pointing out that the paper lacks a clear motivation on why the critical gradients are selected based on the l2 norm. In their response the authors provided a connection with recent methods designed to find important examples when training neural networks. In the discussion (not visible to the authors). Reviewer v5GJ stated that this criterion is intuitive and acceptable given the strong empirical performance reported. The AC agrees with this view.

Reviewer KE2X also points out that the theoretical results do not provide a convincing improvement over vanilla SGD. The authors acknowledge this point, adding that the convergence results apply to a wide variety of critical gradient methods and aggregation strategies, so it is natural to expect that improvements will depend on the specific conditions. While the AC agrees that having stronger theoretical results would improve the paper, as it stands it is certainly above acceptance threshold.

Overall the paper makes a solid contribution with a method that is simple and effective. It will likely inspire other alternative methods in the future. The AC recommends accepting the work.